# Radiomics and Artificial Intelligence in Radiotheranostics: A Review of Applications for Radioligands Targeting Somatostatin Receptors and Prostate-Specific Membrane Antigens

**DOI:** 10.3390/diagnostics14020181

**Published:** 2024-01-14

**Authors:** Elmira Yazdani, Parham Geramifar, Najme Karamzade-Ziarati, Mahdi Sadeghi, Payam Amini, Arman Rahmim

**Affiliations:** 1Medical Physics Department, School of Medicine, Iran University of Medical Sciences, Tehran 14496-14535, Iran; yazdani.el@iums.ac.ir; 2Finetech in Medicine Research Center, Iran University of Medical Sciences, Tehran 14496-14535, Iran; 3Research Center for Nuclear Medicine, Tehran University of Medical Sciences, Tehran 14117-13135, Iran; pgeramifar@tums.ac.ir (P.G.); n.karamzade@gmail.com (N.K.-Z.); 4Department of Biostatistics, School of Public Health, Iran University of Medical Sciences, Tehran 14496-14535, Iran; amini.pay@iums.ac.ir; 5Department of Integrative Oncology, BC Cancer Research Institute, Vancouver, BC V5Z 1L3, Canada; 6Departments of Radiology and Physics, University of British Columbia, Vancouver, BC V5Z 1L3, Canada

**Keywords:** radiotheranostics, radiomics, artificial intelligence, SSTR, PSMA, personalized dosimetry

## Abstract

Radiotheranostics refers to the pairing of radioactive imaging biomarkers with radioactive therapeutic compounds that deliver ionizing radiation. Given the introduction of very promising radiopharmaceuticals, the radiotheranostics approach is creating a novel paradigm in personalized, targeted radionuclide therapies (TRTs), also known as radiopharmaceuticals (RPTs). Radiotherapeutic pairs targeting somatostatin receptors (SSTR) and prostate-specific membrane antigens (PSMA) are increasingly being used to diagnose and treat patients with metastatic neuroendocrine tumors (NETs) and prostate cancer. In parallel, radiomics and artificial intelligence (AI), as important areas in quantitative image analysis, are paving the way for significantly enhanced workflows in diagnostic and theranostic fields, from data and image processing to clinical decision support, improving patient selection, personalized treatment strategies, response prediction, and prognostication. Furthermore, AI has the potential for tremendous effectiveness in patient dosimetry which copes with complex and time-consuming tasks in the RPT workflow. The present work provides a comprehensive overview of radiomics and AI application in radiotheranostics, focusing on pairs of SSTR- or PSMA-targeting radioligands, describing the fundamental concepts and specific imaging/treatment features. Our review includes ligands radiolabeled by ^68^Ga, 18F, ^177^Lu, ^64^Cu, ^90^Y, and 225Ac. Specifically, contributions via radiomics and AI towards improved image acquisition, reconstruction, treatment response, segmentation, restaging, lesion classification, dose prediction, and estimation as well as ongoing developments and future directions are discussed.

## 1. Introduction

Radiotheranostics represents a medical paradigm that uses radiopharmaceuticals for targeted radionuclide therapy (TRT), also known as radiopharmaceutical therapy (RPT). The approach involves the use of the same or different radiopharmaceuticals for both therapeutic and imaging purposes, enabling the matched targeting of specific disease sites [1,2,3]. The radiotheranostics paradigm enables the visualization of drug pharmacokinetics in the body, enabling personalized medicine frameworks [1]. Radiotheranostics makes it feasible to customize treatment planning based on individual variations by choosing “the right drug for the right patient at the right time” [4]. 

This study specifically concentrates on radiotheranostic ligand pairs that selectively bind to somatostatin receptors (SSTRs) and the prostate-specific membrane antigen (PSMA). In general, PSMA expression is higher in prostate cancer (PCa) cells than benign prostate cells, providing a comparatively specific target for patients with this tumor. Moreover, SSTRs are expressed much higher in neuroendocrine tumor (NET) cells or meningiomas than in normal tissues [5].

Prostate cancer is one of the three most common cancers in the world (7.1% of all cancers), with a high survival rate (a 5-year survival rate of >95%) and high recurrence rates [6,7,8,9]. Figure 1A illustrates a simplified disease course for PCa patients. In the first stage, patients are diagnosed through an abnormal serum prostate-specific antigen (PSA) level, a PCa tumor marker. Most of them will be treated (for example, with radiation therapy or surgery), and their PSA will nearly reach zero. However, some of them will have biochemical recurrences. A significant number of PCa patients will progress to metastatic castrate-resistant prostate cancer (mCRPC). Therefore, there is a growing need for alternative therapeutic strategies for these patients. In this regard, several molecules were tested to target PSMA expressed on the cell surface of mCRPC patients [10]. 

PSMA is a type II, 750-amino acid transmembrane protein anchored in the cell membrane of prostate epithelial cells [11]. Radiopharmaceuticals targeting PSMA for diagnostic imaging purposes include [^68^Ga]Ga-PSMA-11, [^68^Ga]Ga-PSMA-617, [^68^Ga]Ga-PSMA-I&T, [^18^F]DCFPyL, [^18^F]PSMA-1007 or [^124^I]MIP-1095, [^64^Cu]Cu-PSMA-617, and [^44^Sc]Sc-PSMA-617. For therapies, [^90^Y]Y-J591, [^177^Lu]Lu-PSMA-617, [^177^Lu]Lu-PSMA-I&T, [^90^Y]Y-PSMA-617, [^225^Ac]Ac-PSMA-I&T, and [^225^Ac]Ac-PSMA-617 have been used [12,13]. Currently, ^68^Ga or ^18^F labeled radioligand binding to PSMA are paramount players in PCa applications [14]. 

In March 2022, [^177^Lu]Lu-PSMA-617 received Food and Drug Administration (FDA) approval as a treatment option for adult patients with PSMA-positive mCRPC, marketed as Pluvicto^®^ [15]. The use of radiolabeled PSMA-targeting ligands provides an important theranostic paradigm, with potential for treating mCRPC patients (Figure 1B) [16,17].

A similar scenario exists for neuroendocrine tumors (Figure 2). Although the epidemiologic importance of NETs is not as high as that of prostate cancer, NETs consist of 0.5% of all malignancies, and the incidence rate has increased six–fold over the past decades [18,19]. Patients with NETs showing high SSTR expression are appropriate candidates for [^68^Ga]Ga-/[^177^Lu]Lu-SSTR applications [20]. 

Several PET-labeled peptides, including [^68^Ga]Ga-DOTA-Tyr3 octreotide ([^68^Ga]Ga-DOTA-TATE), [^68^Ga]Ga-DOTA-Phe1 Tyr3 octreotide ([^68^Ga]Ga-DOTA-TOC), [^68^Ga]Ga-DOTA-1-NaI3 octreotide ([^68^Ga]Ga-DOTA-NOC), and [^64^Cu]Cu bound DOTA-TATE and DOTA-TOC are synthesized for diagnostic applications. Additionally compounds such as Lutetium-177 (^177^Lu), Yttrium-90 (^90^Y), and Actinium-225 (^225^Ac) bound DOTA-TATE and DOTA-TOC, are produced for therapeutic purposes. In January 2018, the FDA approved [^177^Lu]Lu-DOTA-TATE for the treatment of SSTR-positive gastroenteropancreatic neuroendocrine tumors (GEP-NETs) [21]. A complete list of clinically relevant radiotheranostic pairs targeting SSTR and PSMA is shown in Table 1 [22]. 

Artificial intelligence (AI)-based algorithms are increasingly being used to support, simplify, and facilitate dosimetry workflow. Moreover, AI has the potential to predict treatment outcomes and the absorbed dose. Compared to the visual/qualitative assessment of PET images and conventional PET parameters such as the standard uptake value (SUV), radiomics has additional value in diagnostics and prognostics [23]. 

The present review aims to highlight areas of importance in which radiomics and AI can play an essential role in radiotheranostic SSTR- and PSMA-targeting ligand pairs. First, we present a concise review of radiomics and AI. In four sections, we elaborate on the application of radiomics and AI in radiopharmaceuticals targeting SSTR and/or PSMA via image-guided RPTs. Moreover, we briefly explain RPT dosimetry workflows and the role of AI in the dosimetry of ^177^Lu and ^90^Y-labeled-SSTR and PSMA ligand therapies. Finally, we discuss possible future development directions. 

## 2. Radiomics and AI Workflow

In precision medicine, radiomics is currently underway in research based on feature extraction from medical images. Radiomics paves the way to map multimodal imaging into quantitative information on a large scale [24]. Figure 3 depicts the steps required to build a predictive model from medical images [25,26]. The radiomics workflow begins with image acquisition and segmentation. After image post-processing, hundreds of radiomic features (RFs) are measured from segmented regions to provide raw data for developing the final model. The major categories of features are as follows:Geometric or shape features: based on the segmented regions.Statistical or intensity features: computed using intensity values in each image region.Textural features (TFs): quantification of image intensity and regularity via mathematical functions.Wavelet or high-order features: the image transformation process is essential to obtain these features.

Individual features are discarded for dimension reduction through feature selection in the next step. Options include intraclass correlation coefficients (ICC), principal component analysis (PCA), least absolute shrinkage and selection operator (LASSO), recursive feature elimination (RFE), and outputs from machine learning methods (ML).

Different approaches are applied in developing the model depending on the task: univariate or multivariate analysis and supervised or unsupervised ML methods. Supervised ML algorithms are classified into classification and regression algorithms if the variables are categorical or continuous, respectively. Classification algorithms are divided into linear and nonlinear models. Logistic regression and support vector machines (SVMs) are two methods of analyzing linear models. 

In nonlinear models, k-nearest neighbors (KNNs), gradient boosting, decision tree, extra trees (ETs), and random forest are the most commonly used algorithms. The most widely used regression algorithms are linear, logistic, polynomial, support vector regression (SVR), decision regression, random forest regression (RFR), and ridge and lasso regressions.

In contrast to supervised learning methods, unsupervised learning approaches do not contain predefined response variables. Instead, the model finds hidden patterns and insights from the given data. In these procedures, similar data are grouped (clustering), or dimensionality is reduced. Some popular models in this category are K-means clustering, KNN, neural networks (NNs) or artificial neural networks (ANNs), PCA, and independent component analysis (ICA) [27]. 

Deep learning (DL) methods have been introduced as a more comprehensive part of ML methods with various techniques. Classic neural networks, convolutional neural networks (CNNs), recurrent neural networks, auto-encoders, generative adversarial networks (GANs), and gradient descent are examples of DL methods [28]. Challenges still need to be addressed to strengthen radiomics’ role in clinical practice. Most difficulties come from imaging feature variability among different devices and protocols, model robustness, and performance interpretation. In a multicenter context, addressing variability in acquisition and reconstruction protocols is crucial to ensure reproducibility [29]. Accordingly, harmonization procedures have been developed to provide a high reproducibility of RFs in multicenter studies. Reuze et al. [30] and Orlhac et al. [31] reviewed the radiomics workflow and its challenges.

The Society of Nuclear Medicine and Molecular Imaging (SNMMI) AI Task Force published a report on evaluating and validating AI algorithms [32]. This guideline applies extensively to radiomics studies involving AI. According to Figure 4, created based on this guideline, for a prostate cancer patient referred to PET/CT imaging, AI can be applied in a chain from radiochemistry to the physician’s report generation. 

In the first step, AI could predict drug-target interactions, predict and optimize radiochemical reactions, carry out de novo drug design, and optimize radiopharmacy workflows. In the next step, ML-based methods may be well suited to difficult issues in image acquisition and instrumentation. For image reconstruction, AI may offer faster image reconstruction, a better signal-to-noise ratio, and fewer artifacts. 

Image analysis can be automated using AI for different tasks, such as lesion detection, segmentation, and quantification for diagnosis and dosimetry. Moreover, AI has the potential to investigate patterns associated with patient results within large biological and imaging datasets. Additionally, AI can also detect and diagnose. By using ML methods, diagnostic images can be interpreted and translated into reports and clinical databases. Finally, clinicians can receive actionable advice after extracting, distilling, and integrating clinical information from various sources.

## 3. Application of Radiomics and AI in ^68^GA SSTR and PSMA Image-Guided RPTS

The Ga-68/Lu-177 radiotheranostic pair can be used with SSTR and PSMA targeting ligands [33,34]. Gallium-68 (^68^Ga), with a half-life of 68 min and β^+^ emission (89%, *E*_β+max_ = 1920 keV), is suitable for chemical bonding with various chelators. Moreover, it is an appropriate radionuclide for PET imaging of different targets in various diseases. Among these, the most famous are somatostatin receptors in NETs and PSMA in PCa [35,36]. 

The other radionuclide, Lu-177, has a physical half-life of 6.65 days, emitting short-ranged β-rays (*E*_β-max_ = 0.497 MeV; R_ave_ = 0.23 mm in soft tissues) suitable for destroying targeted cancerous cells. Furthermore, it contains γ-rays (*E*_γ_ = 113 and 208 keV; abundance of 6% and 11%, respectively), which help track in vivo radiopharmaceuticals using post-therapeutic imaging [37]. 

In managing patients with mCRPC and NETs who received [^177^Lu]Lu-SSTR peptide receptor radionuclide therapy (PRRT) and PSMA radioligand therapy (RLT), respectively (both fall under the broad umbrella of “RPT” as denoted in most of what follows), the potential of [^68^Ga]Ga-SSTR and PSMA PET RFs has been investigated for different tasks.

Radiomic features can identify patterns and provide additional information not perceptible by the human eyes [38]. In this regard, therapy response assessment, restaging, segmentation, and dose prediction were studied, as presented in the following section. Moreover, Table 2 and Table 3 summarize the radiomics studies on [^68^Ga]Ga-SSTR and PSMA PET imaging for patients who underwent [^177^Lu]Lu-SSTR and PSMA RPT, respectively.

### 3.1. RPT Response Assessment

Therapy response prediction is critical in managing cancerous patients, including early response and biochemical response assessment, recurrence prediction, overall survival (OS), and progression-free survival (PFS). These areas significantly guide clinicians in optimizing treatment strategies during the disease course [39,40,41]. In particular, several studies have investigated the role of RFs in baseline pre-therapeutic [^68^Ga]Ga-SSTR and PSMA PET/CT or PET/MRI to predict patient response to [^177^Lu]Lu-SSTR and PSMA RPT. In some cases, ML methods are applied to different combinations of RFs and clinical parameters to predict therapeutic response.

#### 3.1.1. ^68^Ga/^177^Lu-SSTR

In a multicentric cohort, Werner et al. [65] evaluated the prognostic value of RFs extracted from [^68^Ga]Ga-DOTA-TOC PET/CT before RPT. The authors showed that four heterogeneity features significantly outperform conventional PET parameters in distinguishing responders from non-responders. Moreover, the authors found that “entropy” correlates independently with post-therapeutic PFS and OS in patients who underwent [^177^Lu]Lu-SSTR-analogues, while skewness correlates directly with OS. Furthermore, conventional PET parameters failed to predict these outcomes (see row 1 in Table 3).

In a later study, the same researchers, in a smaller, more homogenous pancreatic NETs (pNET) cohort of patients, found that an intratumoral textural features (TF) analysis of a baseline ^68^Ga-SSTR PET has prognostic value in pNET patients undergoing PRRT [66]. Based on the receiver operating characteristic curve (ROC) analysis, conventional PET parameters like SUVs failed to predict patient outcomes, demonstrating the need for alternative predictors. The results indicated 71% accuracy for entropy values predicting OS. The higher the entropy (with a cut-off > 6.7), the longer the survival (area under the curve (AUC = 0.71) (see row 2 in Table 3) [66].

Önner et al. [67] showed that the evaluation of tumor heterogeneity using two parameters on pretreatment [^68^Ga]Ga-DOTA-TATE PET/CT, namely, skewness and kurtosis, can predict the response of patients with GEP NETs to [^177^Lu]Lu-DOTA-TATE treatment. The researchers reported that these two features were significantly higher in non-responder patients (see row 3 in Table 3).

In 2020, Weber et al. [68] statistically examined the role of a subset of TFs derived from pre-therapeutic [^68^Ga]Ga-DOTA-TOC PET/MRI and apparent diffusion coefficient (ADC) maps to predict PRRT response. The lesion volume on ADC maps and the entropy of the lesions both decreased significantly in the responder patients. No parameters extracted from PET or ADC maps could predict PRRT response. However, these results should be interpreted cautiously due to the small sample size with different treatments (PRRT and conventional therapies) (see row 4 in Table 3).

Ortega et al. [69] utilized parameters of baseline [^68^Ga]Ga-DOTA-TATE PET/CT to predict PFS and the treatment response of patients with NETs who received PRRT. Additionally, an interim PET scan was obtained before the second therapy cycle. The authors used several metrics in their assessment to measure tumor heterogeneity and SSTR expression level, which demonstrated predictive capabilities for PFS. However, changes in these parameters after the first cycle of PRRT did not align with clinical results (see row 5 in Table 3).

Atkinson et al. [70] conducted a pilot study to evaluate the role of texture analysis (TA) applied to baseline [^68^Ga]Ga-DOTA-TATE PET/CT regarding the prognostic potential of tumor heterogeneity and pharmaceutical avidity in patients with NETs who received [^177^Lu]Lu-DOTA-TATE RPT. As a result, tumor textural heterogeneity correlated with shorter PFS. Moreover, kurtosis, skewness, and entropy values derived from PET TA were all positively correlated with survival rates. In the univariate analysis, a larger [^68^Ga]Ga-DOTA-TATE PET uptake tumor area (a newly proposed term by the authors) was substantially related to poor PFS and OS (see row 6 in Table 3).

Liberini et al. [71] reported a pilot study to predict RPT response using RFs of lesions extracted via the [^68^Ga]Ga-DOTA-TOC PET/CT of two patients with NET. The authors showed that two parameters might be applied to RPT response assessment and prediction: whole-body (WB) total-lesion somatostatin receptor expression (TLSRE_wb-50_) and somatostatin receptor-expressing tumor volume (TV) (SRETV_wb-50_) (see row 7 in Table 3).

Laudicellaet al. [72], in 2022, developed the “theragnomics” (THERAGNOstics +radiOMICS) model, a more robust radiomics model to predict the [^177^Lu]Lu-DOTA-TOC RPT response of metastatic patients with GEP NET. The authors analyzed [^68^Ga]Ga-DOTA-TOC PET/CT images before and after RPT. Their findings proved that the “theragnomics” model was superior to conventional quantitative PET parameters in the [^177^Lu]Lu-DOTA-TOC RPT response prediction of patients with GEP-NET lesions. Moreover, skewness and kurtosis were significantly higher in non-responder patients, similar to Önner et al. [67]. Furthermore, compared to Werner et al. [65], SUV_max_ was not significant in the response prediction of RPT and only marginally significant for distinguishing bone lesions in RPT responders and non-responders (see row 8 in Table 3).

#### 3.1.2. ^68^Ga/^177^Lu-PSMA

In univariable survival analysis, Grubmüller et al. [39] showed that after administering two cycles of PSMA-RPT, total tumor volume (TTV) as a first-order RF on PSMA PET had the potential to assess response in mCRPC patients. There was a significant correlation between TTV change and OS (see row 1 in Table 2).

In another study in 2020, Seifert et al. [60] evaluated the role of TTV from PSMA-PET in mCRPC patients’ OS prognosis before [^177^Lu]Lu-PSMA-617 RPT. For each patient, the authors quantified three parameters: TTV (PSMA-TV), total lesion uptake (PSMA-TLU = PSMA-TV × SUV_mean_), and total lesion quotient (PSMA-TLQ = PSMA-TV/SUV_mean_). The results showed a statistically significant negative correlation between PSMA-TV and OS. Also, OS was better predicted by PSMA-TLQ than PSMA-TV independently (see row 4 in Table 2).

Another retrospective study in 2021 by Widjaja et al. [43] reported that ^68^Ga-PSMA PET/CT pre-therapeutic imaging parameters could predict the early biochemical response in patients who underwent [^177^Lu]Lu-PSMA-617 therapy. In this study, SUV_max_ significantly correlated with a PSA change after two cycles, while neither the PSMA-TV nor WB total lesion (TL) PSMA correlated (see row 5 in Table 2).

Khreish et al. [45] evaluated the PFS prediction outcome of [^177^Lu]Lu-PSMA-617 RPT employing [^68^Ga]Ga-PSMA-11 PET-derived parameters (SUV_peak_ and tumor-to-liver ratio (TLR)). In the univariate analysis, responders with partial remission had significantly longer PFS than non-responders (either stable or progressive disease). Also, the assessment of response to TLR in the multivariable analysis was independently associated with PFS (see row 7 in Table 2).

Rosar et al. [50] explored the role of total viable tumor burden from a [^68^Ga]Ga-PSMA-11 PET/CT scan in OS prediction. The researchers semi-automatically determined total lesion PSMA (TLP) through the segmentation of WB tumor and calculated it as the summed products of the volume and SUV_mean_ of all lesions. As a result, early TLP changes independently predicted OS in mCRPC patients who received [^177^Lu]Lu-PSMA-617 RPT (see row 12 in Table 2).

Gafita et al. [51] showed the tumor sink effect on [^68^Ga]Ga-PSMA-11 PET imaging employing quantitative measurements. The result showed that [^177^Lu]Lu-PSMA-617 RPT candidates with high TTV on pre-therapeutic [^68^Ga]Ga-PSMA-11 PET scan without exceeding organ at risks (OARs) radiation dose limit might benefit from increased RPT activity (see row 13 in Table 2).

Due to the time-consuming WB PSMA-TV calculation obtained from PSMA PET scans, in 2022, Hartrampf et al. [52] considered only a limited number of representative lesions. The study showcased the feasibility of RPT response assessment, using the PSMA-TV and SUV_max,_ (of fewer tumor lesions than usual) indicators (see row 14 in Table 2).

Pathmanandav et al. [53] utilized clinical, blood-sample, and imaging biomarkers to report the final safety and efficacy results of a phase I/II study. The study was the combination of [^177^Lu]Lu-PSMA-617 RPT and a radio-sensitizer called idronoxil (NOX66) in mCRPC patients. The results showed that PSMA SUV_max_ was not a treatment response predictor. In contrast, PSMA SUV_mean_, PSMA-avid TV, and treatment duration with an androgen signaling inhibitor were independently correlated with treatment response outcome (see row 15 in Table 2).

In 2018, Khurshid et al. [40], for the first time, examined the potential of tumor textural heterogeneity RFs from pre-therapeutic [^68^Ga]Ga-PSMA PET for [^177^Lu]Lu-PSMA RPT response prediction. Their findings indicated a correlation between increasing PSMA heterogeneity and an enhanced response to PSMA RPT. This contributes to better patient selection, treatment planning, and improved diagnosis (see row 2 in Table 2).

By employing CT texture analysis and ML technique (Weighted KNN algorithm), Acar et al. [41] accurately distinguished metastatic and thoroughly responded lesions in patients imaged through [^68^Ga]Ga-PSMA-11 PET/CT with previous treatment (chemotherapy, radiotherapy, hormonotherapy, [^177^Lu]Lu-PSMA RPT) and known bone metastases. The authors reported that GLZLM_SZHGE and histogram-based kurtosis RFs are imperative in separating metastatic and responding sclerotic lesions (see row 3 in Table 2).

Gafita et al. [44], for the first time, developed nomograms with a combination of clinical and imaging biomarkers via baseline [^68^Ga]Ga-PSMA-11 PET/CT scans to predict [^177^Lu]Lu-PSMA treatment outcome. Nomograms were computed using Cox regression models with the LASSO penalty for parameter selection. The researchers reported that higher PSMA expression was correlated with longer OS and PSA-PFS. Moreover, their nomograms showed that the bone disease was controlled adequately with ¹⁷⁷Lu with a limited chance; patients suffering from bone disease had shorter OS and PSA-PFS (see row 6 in Table 2).

As a proof-of-principle, Moazemi et al. [46] conducted a study using RFs on [^68^Ga]Ga-PSMA PET/CT and clinical parameters to examine their correlation with the difference in prostate-specific antigen levels (ΔPSA) in pre- and post-therapy using linear regression. Moreover, the authors employed the ML approach to predict the treatment response of mCRPC patients who received [^177^Lu]Lu-PSMA-617 RPT and divided them into responders and non-responders. The authors proposed the most effective correlating sets of RFs and clinical parameters with PSA level differences. These sets were further used as surrogate markers for treatment response analyses. Applying ML classifiers to the response prediction task showed that RFs were superior to clinical parameters in correlation with the ΔPSA (see row 8 in Table 2).

Moazemi et al. [47] developed a fully automated clinical decision support tool based on DL methods for mCRPC patients who underwent [^177^Lu]Lu-PSMA RPT. The researchers used a multi-channel UNet to segment 2067 pathological hotspots automatically. Moreover, the authors predicted the response of [^177^Lu]Lu-PSMA RPT based on RFs in [^68^Ga]Ga-PSMA-PET/CT using supervised ML methods. The authors applied the RFE technique to the classification problem to identify the most relevant features. As a result, 14 features were selected. For both automated segmentation and responder prediction tasks, significant results were achieved. Therefore, the results showed that the facilitated automatic decision support tool had the potential to screen mCRPC patients under the RPT (see row 9 in Table 2).

In another study by Moazemi et al. [48], OS prediction in mCRPC patients scheduled for [^177^Lu]Lu-PSMA RPT was investigated. The authors employed RFs from [^68^Ga]Ga-PSMA PET/CT imaging and patient-specific clinical parameters from 2070 delineated hotspots. Using a LASSO regression feature selection method, the results showed that PET kurtosis and SUV_min_ were significantly correlated with OS (see row 10 in Table 2).

As initial evidence, Roll et al. [49] analyzed the predictive and prognostic value of RFs from pre-therapeutic [^68^Ga]Ga-PSMA-11 PET-MRI imaging in 21 mCRPC patients who underwent [^177^Lu]Lu-PSMA RPT. After feature selection, the ten most significant independent RFs discriminated responders from non-responders. Moreover, for biochemical response prediction after RPT, the logistic regression model revealed the highest accuracy. Furthermore, patients with a biochemical response and higher values of T_2_ interquartile range in their PSMA PET imaging showed significantly longer OS (see row 11 in Table 2).

### 3.2. Restaging

Even though [^68^Ga]Ga-SSTR and PSMA PET/CT derived RFs have proven to be a noninvasive tool in primary staging to classify tumors into different groups, some studies have investigated their pivotal role in the restaging of high-risk patients, leading to enhancing the accuracy of the recurrence detection and discrimination of malignancy, and providing profound prognostic information. Moreover, ML, along with RFs, may predict disease progression.

Giesel et al. [54], in 2017, showed that CT lymph node (LN) density measurements correlated with SUV_max_ in [^68^Ga]Ga-DOTA-TOC, [^68^Ga]Ga-PSMA-11, and F-fluorodeoxyglucose (FDG)-PET. The authors found a 7.5 HU threshold to differentiate between malignant and benign infiltration and a 20 HU threshold to exclude benign LN (see rows 16 and 9 in Table 2 and Table 3, respectively). Moazemi et al. [55] employed five ML algorithms on RFs extracted from 2419 hotspots of [^68^Ga]Ga-PSMA-11 PET/CT and classified them as benign (physiologic) or malignant (pathologic) with the same accuracy as the human reader. The authors achieved better performance using PET and CT than PET or CT alone (see row 17 in Table 2).

In 2021, Erle et al. [56] reported that the radiomics decision-tree classification algorithm has a suitable accuracy in classifying the M and N staging of 2452 hotspots of PCa patients on the [^68^Ga]Ga-PSMA-11 PET/CT. The authors showed that combining manual and automated diagnosis has the potential to predict hotspot labels with high sensitivity. However, the liver, kidneys, genitourinary (GU) tract, lacrimal, and salivary glands are the sites where their algorithm had poor performance with a high percentage of false positives (see row 18 in Table 2).

In mCRPC patients, as the naked eye could not detect metastatic bone disease, Hinzpeter et al. [57], as a proof of concept, investigated the role of CT RFs from [^68^Ga]Ga-PSMA-11 PET/CT in bone metastases discrimination. A gradient-boosted tree was trained on the 11 most prominent selected features by employing multi-step dimension reduction and feature selection. The results demonstrated a significant improvement in differentiating unaffected bone from metastatic bone (see row 1 in Table 2).

### 3.3. Segmentation

The accurate detection and segmentation of lesions on [^68^Ga]Ga-SSTR and PSMA PET/CT images is a prerequisite step for individual-treatment planning with [^177^Lu]Lu-SSTR and PSMA to optimize treatment outcomes. Typically, patients with mCRPC and NETs present with an advanced stage of the disease, characterized by a significant number of metastatic lesions distributed throughout their body. Therefore, manual segmentation is not a practical solution in clinical practice because it takes time and effort. In this regard, several authors developed semi-automatic or automatic segmentation methods [38]. The semi-automatic segmentation strategy uses [^68^Ga]Ga-PSMA-SSTR and PSMA PET/CT imaging biomarkers.

Hammes et al. [58], in 2018, proposed a software tool called EBONI for the evaluation of bone involvement that semi-automatically quantifies bone metastasis in [^68^Ga]Ga-PSMA-11 PET/CT. Their software tool produced results in a rapid (3 min/scan), robust, and reproducible way. The results indicated a high correlation between the visual and automatic quantification of bone lesions (see row 20 in Table 2). Zhao et al. [59] developed a triple-combined 2.5D U-NET architecture to detect and segment disease sites automatically on [^68^Ga]Ga-PSMA-11 PET/CT images. The researchers reported that their proposed network achieved higher accuracy in segmenting bone and lymph node metastases (LNM) than local lesions detection (see row 21 in Table 2).

Seifert et al. [42], in 2020, developed and evaluated a software tool to quantify [^68^Ga]Ga-PSMA-11 PET/CT biomarkers semi-automatically. The authors applied percental thresholding of PSMA foci in their proposed software; a SUV_max_ 50% for each focus was used automatically. Additionally, a NN was utilized to semi-automatically exclude physiologic PSMA foci. Moreover, the results indicated that among PSMA PET biomarkers, PSMA_TV_50_ was reproducible and quantified easily with the proposed software. Furthermore, PSMA_TV_50_ significantly predicted OS in patients who received [^177^Lu]Lu-PSMA-617 RPT (see row 22 in Table 2).

In patients with NETs, Liberini et al. [73] extracted RFs from [^68^Ga]Ga-DOTA-TOC PET images and evaluated them according to segmentation methods and intensity discretization. The authors developed semi-automatic edge-based segmentation (SAEB) and applied three fixed SUV_max_ thresholds (20, 30, and 40%). As a result, a SUV_max_ threshold of 40% provides superior RF stability among operators, but biological information may be lost. Due to its superiority over manual segmentation, SAEB segmentation seems to be a promising alternative, but further validation is needed (row 10 in Table 3).

In NETs, hepatic lesions are prominent sites. However, the identification of these sites in [^68^Ga]Ga-DOTA-TATE PET/CT images is challenging due to the high activity of the normal liver background. Wehrend et al. [74] developed a 2D U-Net CNN for the same purpose, automatically detecting hepatic metastases in [^68^Ga]Ga-DOTA-TATE PET/CT images. The authors applied a gradient edge detection method and a pixel noise filter to modify the boundary definition. The researchers showed that the DL algorithm performance was improved when the criteria for lesion boundaries more accurately reflected the true lesion boundary (see row 11 in Table 3).

### 3.4. Dose Prediction

In molecular radiotherapy, dosimetry-based treatment planning is not yet practicable for clinical routines, as it has been performed for external beam radiotherapy. Estimating patient-specific post-therapy dosimetry based on pre-treatment imaging is required to provide a plan before commencing treatment [77,78]. In this regard, different studies show that, for example, pre-treatment [^68^Ga]Ga-SSTR- and [^68^Ga]Ga-PSMA PET/CT imaging is not only used to select appropriate candidates for Lu-177-labeled SSTR and PSMA therapies but also to predict individual post-therapy dosimetry.

Ezziddin et al. [78], in 2012, as a proof of concept, showed that [^68^Ga]Ga-DOTA-TOC PET/CT SUV values (SUV_mean_ or SUV_max_) of tumor lesions in 21 patients significantly correlated with the [^177^Lu]Lu-SSTR AD and treatment response based on serial planar ^177^Lu imaging. Therefore, SSTR PET uptake may predict the therapeutic dose.

The original cohort study by Violet et al. [79] pointed out that SUV values on [^68^Ga]Ga-PSMA PET/CT correlated with the AD of [^177^Lu]Lu-PSMA RPT and PSA response. The results showed that SUV_mean_ correlates better with AD in tumor lesions than SUV_max_. These two initial findings and others [80,81,82,83] should be further validated on more datasets, different therapies, and clinical situations. AI models play a critical role in personalized RPT planning. In this regard, Xue et al. [61] employed the previously developed DoseGAN software, originally designed for stereotactic body radiation therapy (SBRT), to predict voxel-wise absorbed dose in [^177^Lu]Lu-PSMA RLT. This prediction took into consideration pre-therapeutic images from [^68^Ga]Ga-PSMA PET/CT. The authors trained a GAN as a dual-input model based on PET/CT information, utilizing 3D absorbed-dose maps acquired through Hermes Voxel Dosimetry (see row 23 in Table 2). To pre-train their GAN model, they further generated 266 digital phantoms using the extended cardiac-torso or XCAT phantoms, employing a physiologically based pharmacokinetic (PBPK) model for phantom generation [62]. This approach allowed them to simulate diverse PET images and the spatiotemporal distribution of therapy ligands, ultimately improving dose prediction accuracy (see row 24 in Table 2). In a complementary investigation underscoring the significance of 3D-kinetic models for dosimetry, Kassar et al. [84] explored data augmentation with a PBPK model in a conditional GAN or cGAN to improve organ-specific RLT prediction. Using virtual patient data from realistic simulations, they demonstrated significant enhancements in dose accuracy when incorporating the PBPK constraint. The results suggest that aligning network predictions with mechanistic, patient-specific models addresses limitations in DL for personalized RLT treatment planning.

As a proof of concept, Xue et al. [63,64] investigated the feasibility of the individual dose prediction of [^177^Lu]Lu-PSMA-I&T based on pre-therapy [^68^Ga]Ga-PSMA PET/CT imaging employing ML techniques (ANN and RFR). The authors compared the accuracy of their dose prediction results with population-based estimation and found a significant error in the latter. Nevertheless, their findings should be proved for lesions (see row 25 in Table 2).

Recently, Akhavanallaf et al. [75] developed ML models to predict therapeutic tumor dose using pre-therapy ^68^Ga PET and clinicopathological biomarkers for patients with metastatic NETs treated with PRRT. This study retrospectively analyzed 90 segmented metastatic NETs from 25 patients who underwent pre-therapy [^68^Ga]Ga-DOTA-TATE PET/CT and SPECT/CT at four time points after ^177^Lu-DOTA-TATE administration. SUV_mean_, TLSUV_mean_, (SUV_mean_ of total-lesion-burden), and total liver SUV_mean_ were found to be the most reliable predictors of tumor dose. A trivariate RF model combining these metrics provided the highest performance in tumor dose prediction. The study demonstrates the feasibility of using baseline PET images for absorbed dose prediction prior to ^177^Lu-PRRT. It forms the groundwork for ^68^Ga-PET’s role in personalized treatment planning and patient stratification in the era of precision medicine (see row 12 in Table 3).

In a ground-breaking retrospective study, Plachouris et al. [76] employed the power of cutting-edge ML regression algorithms to predict OARs absorbed doses in patients suffering from NETs, sourced from two clinical centers. Their innovative approach involved integrating radiomic features from [^68^Ga]Ga-DOTA-TOC PET/CT scans with dosiomic features extracted from dose maps of [^177^Lu]Lu-DOTATATERPT treatment cycles. These radiodosimetric features have the potential to offer insights into the potential recurrence of any disease and could prove valuable in clinical decision-making, particularly in addressing dose escalation concerns (see row 13 in Table 3).

## 4. Application of Radiomics and AI in [^18^F]PSMA PET/CT Image-Guided RPTS

Pears and pitfalls associated with ^68^Ga compared to Fluorine-18 (^18^F) are comprehended. ^68^Ga intrinsic characteristics contribute to image noise. For example, the positron yield of the ^68^Ga radioisotope is low, which increases the number of counts and adds more noise to a scan. Moreover, ^68^Ga has a significantly higher range and positron energy, increasing noise and contributing to the partial volume effect. This issue can influence small lesions’ detectability [85]. Therefore, in recent years, there has been a demand for PSMA ligand imaging using ^18^F-labeled radiopharmaceuticals instead of ^68^Ga compounds due to their inherent advantages, such as less noise and less urinary bladder activity [85,86,87].

^18^F is the most commonly used radioisotope for PET imaging. ^18^F is produced using a cyclotron with a high positron emission yield (97%), short half-life (109.7 min), and low positron energy (0.635 MeV), resulting in high-resolution images because of the short diffusion range. Due to its longer half-life than ^68^Ga and its ability to be manufactured centrally and delivered to satellite sites, ^18^F is more suitable for commercial applications [85].

The first generation of ^18^F-labeled PSMA ligands is [^18^F]DCFBC. The drawback of this radiopharmaceutical is its high background activity, which is addressed by the second-generation compound [^18^F]DCFPyL. This PSMA ligand is characterized by its fast urinary excretion, which could affect the pelvic lesion’s detectability [85]. Recently, [^18^F]DCFPyL received FDA approval and was marketed as Pylarifly^®^ [88]. It is difficult to visualize metastases adjacent to the prostate gland with this radiopharmaceutical [86]. Furthermore, there is no chelator present in either [^18^F]DCFBC or [^18^F]DCFPyL that can bind therapeutic nuclides. Therefore, at different stages of clinical evaluation, alternative [^18^F]PSMA ligand pharmaceuticals were employed, and all showed impressive image quality, e.g., [^18^F]PSMA-1007, [^18^F]AlF-PSMA-1, and [^18^F]JK-PSMA-7 [85,86]. These ligands have been used for many purposes, such as prostate cancer imaging, initial diagnosis, biochemical recurrence [89], and restaging metastatic lesions [85,86,87,89].

In the case of [^18^F]PSMA-1007, structurally related to PSMA-617 (^18^F is labeled to DKFZ-PSMA-617 initially developed for ^68^Ga-ligand), the liver metabolizes the PSMA-1007 ligand pharmaceutical instead of urinary excretion [87]. Moreover, in a metaphor for the chemical structure, the uptake of tumors and normal organs are very similar to [^18^F]PSMA-1007 and [^177^Lu]Lu-PSMA-617 compounds [90,91]. Therefore, [^18^F]PSMA-1007 can be viewed as a well-suited diagnostic equivalent to PSMA-617, which may help guide the choice of patients referred to PSMA-617 therapy. Also, it can be used for staging and the detection of PCa recurrence.

Additional investigation with large cohorts must be conducted to demonstrate if [^18^F]PSMA-1007 can be used as a theranostics pair for PSMA-617 instead of [^68^Ga]Ga-PSMA-11 in the future [85,87,90,92]. Despite having several advantages over [^68^Ga]Ga-PSMA-11, [^18^F]PSMA-1007 also has a major drawback as occasional unspecific bone uptake [57,93].

PSMA is expressed in prostate tissue but is also found in inflammatory and neovascular tissue. Therefore, activated bone marrow granulocytes and islands, especially in the rips and extremities, which are favored locations for bone metastases from prostate cancer, may be a rationale for UBU. Additionally, UBU has been linked to other bone abnormalities such as fibrous dysplasia or Paget’s disease [94].

One potential areas of research is applying the radiomics concept into the analysis of clinical [^18^F]PSMA-1007 PET/CT images. This strategy will provide a better understanding of the lesion features that accurately depict prostate malignancy and those that show benign uptake. It will also distinguish bone metastases from non-specific PSMA uptake.

### 4.1. RPT Response Assessment

PET-derived assessments of TTV imaging biomarkers are expected to play an increasingly significant role in assessing RPT response in mCRPC patients [95,96]. For the first time, Unterrainer et al. [97], in a pilot study, evaluated baseline or interim TTV in comparison to the clinical course of RPT with [^225^Ac]Ac-PSMA-I&T in thirteen mCRPC patients with available pre-therapeutic ^18^F-PSMA-1007 PET/CT. The authors concluded that most patients qualified for ≥2 cycles of [^225^Ac]Ac-PSMA-RPT exhibited rapid TTV declines that were not correlated directly with changes in other indicators, such as serum PSA. Since TTV reflects the current tumor load without considering the development of newly formed lesions, evaluating changes in TTV may enhance response assessment compared to standard response classifiers like RECIST 1.1 and mPERCIST.

### 4.2. Segmentation

Various approaches and settings for [^18^F]PSMA-PET/CT have been proposed to address tumor delineation issues and obtain the accurate representations of lesions that require functional contours for metabolic quantification in routine clinical practice [98,99].

In a study, Mittlmeier et al. [100] assessed the correlation of different PET-based delineation thresholds on [^18^F]PSMA-1007 PET with CT-based large, non-bulky LNM volume measurements for 50 patients with metastatic prostate cancer. The Shapiro–Wilk test was used to establish whether the data had a normal distribution, and the results were then evaluated using Spearman and Pearson correlation coefficients (CC). Irrespective of potential alterations in PSMA-avidity in background tissues such as parotids, a simple SUV threshold of 4.0 for the delineation of nodal PCa lesions showed the most substantial relationship with the volumetric reference standard.

Lau et al. [101] included 275 lesions in 68 patients diagnosed with biochemical recurrence after total prostatectomy. The authors measured metabolic parameters on [^18^F]PSMA-1007 dual time point PET/CT images using threshold-based and slope-based methods under different acquisition times. Functional contours were obtained, and prostate cancer lesions with minimal uptake time fluctuation were quantified. The authors recommended the gradient-based method with a high completion rate for segmenting and quantifying prostate cancer lesions on [^18^F]PSMA-1007 PET/CT imaging.

Trägårdh et al. [102] developed a freely available, fully automated AI-based method to detect and quantify suspected prostate tumor/local recurrence, lymph node metastases, and bone metastases from [^18^F]PSMA-1007 PET/CT images of 660 patients. A CNN was trained, validated, and tested on 420, 120, and 120 patients’ datasets, respectively. The network results were compared with ground truth segmentation accomplished by several nuclear medicine physicians. The authors assessed tumor burden, including total lesion volume (TLV) and TLU. The average sensitivity of the AI technique was 79% for detecting bone metastases, 79% for lymph node metastases, and 79% for prostate tumor recurrence. The comparable average sensitivities for nuclear medicine physicians were 78%, 78%, and 59%. The TLV and TLU correlations between AI and nuclear medicine physicians ranged from R = 0.53 to R = 0.83 and were all statistically significant.

One of the main areas of research in the future is employing radiomics and/or AI approaches to appropriately delineate and quantify the lesions on pre-therapeutic [^18^F]PSMA-1007 PET/CT of mCRPC patients for different applications such as RPT monitoring, response assessment, and dose prediction. [^18^F]PSMA-1007 has the potential for a drastic impact on precision medicine in the coming years.

## 5. Application of Radiomics and AI in ^64^Cu SSTR and PSMA Image-Guided RPTS

Somatostatin analogues labeled with ^64^Cu have been developed, and a head-to-head comparison of [^64^Cu]Cu-DOTA-TATE and [^68^Ga]Ga-DOTA-TOC revealed the former’s advantages in identifying lesions in patients with NETs [103]. For centers lacking ^68^Ge/^68^Ga generators, ^64^Cu ligands with a half-life of 12.7 h offer practical benefits because these ligands can be efficiently delivered from other production facilities. Moreover, the high target-to-background contrast obtained and the high detection rate of suspected lesions in NETs patients hold promise for the secure administration of [^64^Cu]Cu-DOTA-TOC for NETs patients [104].

^64^Cu has a superior spatial resolution to ^68^Ga due to its lower positron range, which could aid in diagnosing small lesions [103]. Furthermore, a 100% sensitivity and 96.8% specificity were discovered with no side effects in the phase III clinical trial assessing [^64^Cu]Cu-DOTA-TATE PET/CT imaging for NETs [105]. Therefore, considering all of these advantages, the FDA-approved [^64^Cu]Cu-DOTA-TOC (also named Detectnet^®^) can be implemented for pre-therapy dosimetry or any other logistical concerns in the routine clinical setting [103,104].

The increased radiation load of ^64^Cu-DOTA-TATE is a potential issue. The injection of 180–220 Megabecquerel (MBq) per patient results in a radiation dose of 5.8–8.9 mSv, roughly two times more than the standard dosage of [^68^Ga]Ga-DOTA-TOC (120–200 MBq, 2.8–4.6 mSv radiation dose) [103]. Moreover, the inadequate lesion uptake of the radiopharmaceutical at late time points is another drawback of [^64^Cu]Cu-DOTA-TATE PET/CT imaging. The cause could be that DOTA is not the best chelating agent for Cu-64 since copper DOTA complexes are too unstable in acidic environments [106].

PSMA-617 was labeled with ^64^Cu [107] and imaged in first-in-human clinical experiments [108]. There was a high uptake (SUV_max_) in metastatic lesions with low background activity. According to the study’s findings, PET imaging with [^64^Cu]Cu-PSMA-617 has a significant role in the primary staging of some patients and patients with recurrent disease, particularly in centers without access to [^68^Ga]Ga-PSMA ligands. Following the theranostics model, [^64^Cu]Cu-PSMA-617 can be used to choose patients for ^177^Lu RPT. It opens the door to pre-therapeutic radiation dosimetry in the context of patient-specific and individualized RPT in the future [108].

### 5.1. RPT Response Assessment

In two prospective studies, Carlsen et al. [109,110] evaluated 164 patients with neuroendocrine neoplasms (NENs) who underwent [^64^Cu]Cu-DOTA-TATE PET/CT SSTR imaging. According to the first study’s results, the OS and PFS of patients with NETs were estimated using [^64^Cu]Cu-DOTA-TATE PET images. Kaplan–Meier (KM) analysis with log-rank was used to calculate the predictive value of [^64^Cu]Cu-DOTA-TATE SUV_max_ for OS and PFS. Despite not identifying a cut-off to predict OS, the results showed that patients with a tumor SUV_max_ value greater than 43.3 had half the chance of disease progression than those with values of 43.3 or less. The authors reported that, with this cut-off of 43.3 for SUV_max_, PFS could be predicted after 24 months of follow-up with a moderate accuracy of 57%. It is simple to determine a patient’s maximum tumor SUV_max_, but as it represents the highest somatostatin receptor density, the prediction is likely based on the most differentiated and least aggressive tumor area [109].

In a later study, Carlsen et al. [110] developed a standardized semi-automated tumor delineation approach to find the lesion with the lowest uptake. Moreover, the authors also evaluated the TTV obtained from the semiautomatic tumor delineation. If there was any association between OS and PFS, it was found using the KM and Cox regression methods. Compared to the previous technique based on SUV_max_ [109], ^64^Cu]Cu-DOTA-TATE PET/CT significantly improved the prognostic value by assessing the lowest lesion uptake rather than the highest. Therefore, an improved prognostic classification method for patients with NENs was created by combining lesion uptake and TTV [110]. In these two studies, there was no specific matching comparison of the prognostic significance of ^68^Ga-labeled imaging in the same population. However, it was previously evident that SUV_max_ was higher for [^64^Cu]Cu-DOTA-TATE than [^68^Ga]Ga-DOTATATE in liver lesions, lymph nodes, pancreatic lesions, intestinal tumors, and carcinomatosis lesions [103].

### 5.2. Segmentation

In 2022, Carlsen et al. [111] developed a U-net architecture using the nnU-Net framework for the tumor segmentation of NENs in [^64^Cu]Cu-DOTA-TATE PET. Among three [^64^Cu]Cu-DOTA-TATE datasets (including 127, 31, and 10 PET/CT images), a randomly selected subset of 117 from dataset (I) was implemented for training and validation. The test cohort comprised 31 patients from dataset (II) and the remaining ten from dataset (I). Patients from datasets (I) and (II) were tested to account for any potential effects of using various PET/CT systems. The authors also tested the AI model on patients from dataset (III) or patients who had radical surgery and showed no evidence of NEN on PET/CT. The volume and number of segmentations (if any) were used to evaluate these patients. A standardized semi-automatic method for tumor segmentation by a physician produced ground truth segmentations. The proposed AI model had a pixel and lesion-wise dice score of 0.850 and 0.801 in the test cohort, without manual adjustments. As a result, their approach produced substantially faster tumor segmentation while maintaining high concordance with ground truth segmentation. Total tumor segmentation may become more practical in daily clinical practice with AI.

## 6. Dosimetry Workflow and Treatment Planning

In current RPT practice, a fixed-dose activity administration based on patient characteristics has been used. This strategy eventually leads to an under-dosage of tumoral lesions and over-dosage of OARs because the essential rule of patient-specific characteristics, such as anatomical and functional variations, not to mention variations in the radiation beam properties, is not considered [112].

The most promising remedy to overcome these limitations is integrating dosimetry-guided treatment planning into RPT practice [113]. Dosimetry-based treatment takes rational decisions for dose delivery based on earlier variations to maximize tumor AD while preventing toxicity to critical organs [114]. Therapeutic radionuclides such as Y-90, Lu-177, and Ac-225 can be imaged quantitatively, enabling personalized dosimetry calculations [115,116,117].

The dosimetry workflow, from image acquisition to absorbed dose calculation, is shown in Figure 5A. The workflow begins with accurate scanner calibration and the accurate measurement of administered activity to determine a calibration factor. Serial image acquisitions are acquired at least at three-time points in order to measure the activity distribution in lesions and target organs. Three imaging protocols are proposed for dosimetry based on scanner availability: whole-body 2D or planar, 3D, and 2.5D or hybrid acquisitions, as shown in Figure 5B.

Medical Internal Radiation Dose (MIRD) pamphlet 16 provides general recommendations for optimal imaging time points [118]. MIRD pamphlet 23 guides SPECT quantification based on patient-specific dosimetry [119]. Moreover, MIRD pamphlet 26 considers ^177^Lu SPECT quantification in RPT dosimetry [120].

The next step is image correction for adverse effects, such as artifacts, distortions, and noise. Time-integrated activity (TIA) or cumulative activity is essential for dosimetry in the next step. Analytical methods are applied to plot the time–activity curve (TAC) of activities related to the first to last time points. The TIA for each delineated tissue is estimated via the TAC time integral [22]. The last step is the absorbed dose calculation to convert cumulative activities into absorbed doses. In this regard, there are two principal methodologies: (I) organ or phantom-based, and (II) voxel or patient-based.

The organ or phantom-based dosimetry approach is based on a formalism provided by the MIRD committee that calculates the mean AD of target-source organs per radioactive decay. This approach employs computational phantoms to model the TIAs and physical features of the radionuclides, called S-values [121,122]. These models consist of three approaches: local energy deposition (LED); dose point kernel, voxel S-value; and Monte Carlo (MC) simulation (Figure 5C).

The gold standard and the most accurate personalized dosimetry method is the direct MC simulation of radiation transport. This can consider both heterogeneous activities and medium distributions. However, extensive computational time, effort, and resources provide this level of accuracy, precluding its application in clinical routines [22,123].

Although a magic bullet for fast and accurate internal dosimetry is not recommended, researchers are progressing slowly. Recently, the EANM committee provided recommendations for the dosimetry of [^177^Lu]Lu-SSTR and [^177^Lu]Lu-PSMA [5].

## 7. Role of AI in Dosimetry Workflow of ^177^Lu-SSTR and PSMA RPT

AI has the potential to be used in the internal dosimetry workflow to make it more efficient, qualified, reproducible, and feasible in daily clinical practice [124]. In this regard, AI can be applied to different dosimetry steps: image acquisition, image quantification, image registration, image segmentation, kinetic modeling, dose assessment, and dose prediction, to name but a few. The following is a summary of studies that applied AI in different steps of [^177^Lu]Lu-SSTR and [^177^Lu]Lu-PSMARPT dosimetry procedures.

### 7.1. Image Acquisition and Quantification

AI-based image quantification algorithms are increasing gradually for different modalities and applications to ease clinical decision-making [125]. In the dosimetry practice, the quantification process has the potential to be enhanced by AI. Serial PET or SPECT acquisitions of 2–3 bed positions to cover the critical organs with attenuation, scatter, and collimator–detector response corrected ordered subset expectation maximization (OSEM) reconstructions are necessary for image quantification and for enabling accurate ^177^Lu kinetics. This step is complicated, time-consuming, and modality-dependent. It can take 30 to 60 min for each time point. However, the available camera time is limited and requires restricted acquisition times per bed position for patient comfort, reducing motion artifacts and image noise [126,127].

AI is an ideal tool to streamline this step, according to a recent study by Rydén et al. [128], which successfully reduced SPECT/CT acquisition time by reducing projection numbers. In this effort, the authors developed a U-shaped CNN to generate synthetic intermediate projections (SIPs) of [^177^Lu]Lu-DOTA-TATE SPECT to avoid image degradation. A total of 352, 37, and 15 SPECT images, each consisting of 30 projections (selected as every fourth projection out of 120), were employed for the training, validation, and testing of three separate CNN models. During the training phase, the researchers utilized In-111 and Lu-177 data, whereas for the testing phase, only Lu-177 data were employed.

The output was 30 SIPs. The authors evaluated their network through raw data of SPECT/CT images from both a Jaszczak cylinder phantom with six hot spheres and 15 patients treated with [^177^Lu]Lu-DOTA-TATE. Activity concentrations of kidneys were determined through different SPECT images to compare. The results indicated that kidney activity concentration is comparable using different projection sets. The statistical results revealed that the quality of SPECT images was improved by adding SIPs to sparsely sampled projection data. Their proposed approach reduced scan-time duration on the one hand and avoided image degradation on the other.

### 7.2. Image Segmentation

The most challenging task in dosimetry workflow is the manual segmentation of OARs and tumors. The estimated absorbed radiation dose of delineated OARs and tumors highly depends on segmentation accuracy. Manual segmentation methods such as threshold, shape-based, watershed, and region-grow are error-prone, time-consuming, operator-dependent, and susceptible to intra- [129,130] and inter-operator variability [131,132].

The SNMMI Dosimetry Task Force “challenge” conducted an international project to identify, understand, and characterize variations in dosimetry workflow [133]. For the same imaging data from two patients administered with [^177^Lu]Lu-DOTA-TATE, different radiation dose estimates were reported from different centers worldwide, indicating variations in methods used in each dosimetry step. Variations in curve fitting and segmentation of the region of interest (ROI) or volume of interest (VOI) play crucial roles [133].

Automated tumor segmentation has emerged in most research areas to address all the shortcomings of manual or semiautomatic segmentation procedures [38]. A review by Brosch-Lenz et al. [124] discussed prominent and emerging segmentation methods and their possible applications in RPT dosimetry. The following are just a few studies that applied DL segmentation algorithms accurately and relatively fast to [^177^Lu]Lu-PSMA or [^177^Lu]Lu-SSTR dosimetry.

Jackson et al. [134] showed how to detect and segment kidneys automatically on non-contrast CT images with a 3D U-Net model. The authors trained a model on 89 manually segmented cases and tested it on patients who underwent [^177^Lu]Lu-PSMA-617 RPT. The authors achieved accurate contours in around 90 s with mean dice coefficients of 0.91 and 0.86 for the left and right kidneys, respectively. Although results for three patients with cystic regions in their kidneys showed poor performance, their manual and automated algorithms revealed moderately similar mean radiation AD results.

Another proposed study on automated kidney segmentation by Ryden et al. [135] aimed at the dosimetry of Lu-177-based SPECT/CT images. The researchers developed a 3-D UNet dual-input model trained on 119 SPECT/CT images and validated on 13 images. Their results showed that incorporating both SPECT and CT is necessary to obtain accurate and precise segmentation. SPECT quantification failings are sometimes due to organ movements between SPECT and CT images. The results of both studies were promising; however, the application of 3D DL algorithms, which are computationally expensive and not suitable for daily clinical practice, should be considered.

Nazari et al. resolved this issue [136] by developing a modified 2.5D CNN model to automatically delineate the organ borders using post-treatment CT images of patients treated with [^177^Lu]Lu-DOTATOC. The authors developed a CNN model based on the mask-rcnn structure, capable of segmenting kidneys and liver, with dice scores of 94% and 93%, respectively. Moreover, the authors estimated the kidney dosimetry results of eight patients with automated segmentation. The results showed a 7% difference from manual segmentation by two medical physicists.

### 7.3. Dose Estimation

MC simulation, as the gold standard for voxel-level patient-specific dosimetry, suffers from extensive computational time and resources, so its application is limited in daily clinical practice [137]. AI is one of the first proposed solutions to improve patient-specific dosimetry accuracy and computational efficacy.

Lately, DL methods have been employed to generate accurate 3D absorbed dose maps or dose rate maps with a considerable reduction in computation time. In light of this idea, Lee et al. [138] were the first group to prove the feasibility of the DL method in estimating the voxel dose. To assess angiogenesis, the authors applied U-Net deep neural architecture to the 3D PET/CT image patch of ^68^Ga-NOTA-RGD with the matrix and voxel sizes of 48 × 48 × 24 and 2.6 × 2.67 × 5 mm^3^, respectively. Their results showed comparable results with MC simulation but significantly lower computation time.

Gotz et al. [139] applied a hybrid modified U-Net/empirical mode decomposition method to estimate individual dose rate maps for patients who received the [^177^Lu]Lu-PSMA RPT. The authors used SPECT and CT images as dual inputs for training to provide a MIRD-based voxelized dose map and a patient-specific tissue density map, respectively.

Melodia et al. [140] employed CNN to estimate dose voxel kernels (DVKs) using CT density maps for the first time. The authors verified their network on an actual patient SPECT/CT and assumed that the decay distribution was known. Gotz et al. [141] applied the same approach and trained a NN with MC simulations of WB CT as inputs to predict DVK for kidney dosimetry of 26 patients undergoing therapy with [^177^Lu]Lu-PSMA or [^177^Lu]Lu-DOTA-TOC.

Most of the mentioned works feed their networks using MC simulation to provide ground-truth data. Although this simulation was performed only once, providing a full-scale training dataset is challenging due to expensive MC computation. In this regard, Ahavanallaf et al. [142] extended the voxel-wise MIRD strategy from a single S-value kernel into specific S-value kernels based on patient-specific anatomy. The authors predicted the distribution of deposited energy in the whole-body organ-level dosimetry of [^18^F]FDG PET imaging. Also, the authors compared their network performance against direct MC practice. It should also be mentioned that the proposed strategy has not, thus far, been implemented for RLT or PRRT dosimetry.

One of the network training limitations in the mentioned studies is the inaccurate ground truth dose distributions derived from SPECT-based activity maps because of a poor SPECT imaging resolution. Li et al. [143] trained and tested a deep residual CNN using MC-generated dose-rate maps of virtual patient phantoms to overcome this issue and generate reliable dose-rate maps. The phantoms corresponded to the activity/density maps of 13 patients who underwent [^68^Ga]Ga-DOTA-TATE PET/CT scans. Moreover, the authors verified their network on 42 SPECT/CT scans of patients who underwent [^177^Lu]Lu -DOTA-TATE. Compared to DVK and MC approaches, their model called “DblurDoseNet” improved accuracy, resolution, noise reduction, and speed across multiple regions (kidneys, lumbar vertebra, and lesions in soft tissues and lung), making it possible to be used in clinical treatment planning.

To overcome the computational burden of MC simulations, Dewaraja et al. developed an automated voxel dosimetry pipeline. Essential steps in their PRRT dosimetry workflow included organ (but not tumor) segmentation, registration, dose-rate estimation, and curve fitting, which were automated using an integrated workflow. Using CNN, organs on CT were automatically segmented through SPECT/CT images. The authors employed the fast dose-planning method MC code to estimate the dose rate using explicit MC. A function was selected automatically for each voxel, which was more optimal to fit the dose rate. Data from four time-point SPECT/CT images of 20 patients with 77 NETs, who underwent [^177^Lu]Lu-SSTR PRRT, was utilized.

The entire automated dosimetry, including CNN organ segmentation, co-registration, MC dosimetry, and voxel curve fitting, was processed on a desktop computer in 25 min. Their state-of-the-art tool could be implemented in clinical dosimetry-guided RPTs. To establish generalizability, more training/testing datasets are needed.

## 8. Role of AI in Dosimetry Workflow of ^90^Y SSTR and PSMA RPT

Yttrium-90 (^90^Y), a theranostic agent with a physical half-life of 64.1 h, can be used in PRRT and RLT [116,144,145]. The pure emission of high-energy β^−^ particles from ^90^Y produces a continuous spectrum of Bremsstrahlung radiation. In dosimetry, Bremsstrahlung emissions or very low positron emissions can quantify ^90^Y. Therefore, the post-administration activity quantification of ^90^Y becomes challenging due to internal photon scattering and high additional background from the Bremsstrahlung radioactivity [146].

There is a growing concern and effort in developing specialized reconstruction techniques to optimize ^90^Y quantification and promote its accuracy. In this regard, methods based on MC simulation modeling [146,147] and NNs [148,149,150] have been proposed, although the latter is preferred due to computationally intensive MC-based simulations. Using AI for SPECT image reconstruction was explored and showed better quantification results than conventional SPECT reconstruction methods, such as OSEM [151].

Xiang et al. [148] developed a deep CNN (DCNN) with 13 layers for fast scattering projection estimation from ^90^Y SPECT/CT images. The training dataset was from high-count MC-simulated anthropomorphic and non-anthropomorphic digital phantom data. The three testing data included a simulated sphere phantom, a liver phantom measurement, and patients. The estimated scatter projections were fed into the OSEM reconstruction algorithm to compensate for scattering [148]. Regarding accuracy and computation time, the DCNN for test data that involved patient investigations was in reasonable agreement with MC simulation results at a fraction of the time. Using a single desktop processor, the DCNN generated a patient scatter projection for 128 views in 1 min. This was approximately three orders of magnitude faster than MC’s Bremsstrahlung scattering estimation.

## 9. Clinical Perspectives on Radiomics and AI

The complexity of patient-specific dosimetry poses challenges for implementation and puts a substantial burden on medical physicists, technologists, and physicians. A prerequisite for personalizing RPTs is not only accuracy and reliability but also practicality. Routine development of personalized medicine will become more likely through developments that automate, simplify, or accelerate the dosimetry workflow steps. A key component of personalized RPT in the future will involve the use of radiomics and AI methods in the field of radiotheranostics. Personalized therapies can be easily implemented in clinical settings with the assistance of emerging AI research and applications in quantitative imaging, segmentation, absorbed dose estimation, absorbed dose prediction, and outcome modeling. Nevertheless, this transition is not without its challenges, and the implementation of personalized RPTs requires careful consideration of limitations and uncertainties [152].

Currently, patient-specific RPTs are not routinely implemented in clinical settings [113]. In recent years, the emergence of radiomics and AI has provided a means to streamline these tasks, with the possibility of integrating them into clinical practice. In this work, possible applications of radiomics and AI in the clinical radiotheranostics scenario are highlighted. The intention is to inspire the community to broaden and align efforts to achieve routine and reliable RPTs personalization. In this regard, comprehensive validation and fine-tuning on larger patient cohorts are necessary to refine personalized RPT strategies and facilitate clinical translation. Furthermore, radiomics and AI applications in clinical practice remain challenging and remain to emerge. Image reconstruction algorithms, gray-level intensity discretization, and tumor segmentation methods all influence the measurement of quantitative imaging biomarkers [26,153]. These factors may affect the robustness, repeatability, and reproducibility of the variables and their outcomes. As such, there is a need to increase the robustness of these tools. For instance, the radiomics quality score (RQS) and the imaging biomarker standardization initiative (IBSI) have been introduced to enhance radiomics robustness, offering methodological guidance and standardization for high-throughput image analysis [26,154]. By improving the understanding of the technical aspects of radiomics and AI, these instruments contribute to gradual harmonization and standardization. With this advancement, radiomics and AI will have more tangible and less hypothetical applications in clinical settings in the future.

## 10. Discussion and Future Directions

Recent advances in AI have gained considerable attention in the nuclear medicine field. Recently, the SNMMI AI task force published “best practices for algorithm development” and “best practices for evaluation (the RELIANCE guidelines)” and provided recommendations for developing AI algorithms in nuclear medicine, likewise principally concerned radiomic investigations [32,155].

Saboury et al. [156] proposed a roadmap toward trustworthy AI ecosystems in nuclear medicine. The authors referred to the twelve key elements of trustworthiness in AI systems entitled “human agency”, “oversight”, “technical robustness”, “safety”, “privacy”, “predetermined change control plan”, “security”, “diversity and bias awareness”, “stakeholder participation”, “transparency and explainability”, “sustainability of societal well-being”, and the last but not least, “fairness and supportive context of implementation”. In massive imaging or biological datasets associated with patient results, modern AI methods uncover hidden but meaningful patterns. AI can, therefore, supplement radiomics [157].

The traditional radiomics methodology extracts hand-crafted or engineered features from a segmented ROI or VOI. However, DL’s recent advancements have led to a trend for DL-based radiomics. In light of the advantages of these two methods, hybrid solutions were created to take advantage of different data sources. Moreover, ML algorithms may be employed for mining large quantities of radiomic features. These features may be augmented with additional clinical data or omics to find associations, eliminate redundancy, build tractable representations in low-dimension spaces, or develop prediction models. Furthermore, unsupervised ML can be used to select pertinent features for a task or combine associated input features into a more manageable set of elements [32,157].

To the best of our knowledge, this paper is the first review of the current applications of radiomics and AI methods, in both pre-therapeutic radiolabeled-SSTR and PSMA PET/CT or PET/MR images for different tasks. Our primary focus is on studies that extracted image parameters from radiolabeled-SSTR and PSMA PET/CT images for different tasks, such as predicting treatment response, restaging, segmentation, and dose prediction using various analysis methods, including univariate and multivariate statistical analyses and ML approaches.

This paper also addresses the application of DL in PRRT and RLT dosimetry. The OS of patients with NETs and mCRPC treated with [^177^Lu]Lu-SSTR and PSMA RPTs was significantly improved. However, this improvement was not observed in all patients [15,21]. As such, one of the underlying approaches to providing better performance for all patients undergoing RPTs is personalized treatment plans, which are made possible through precise and reliable dosimetry calculations. At the same time, the dosimetry workflow for RPT needs to be improved and facilitated in routine clinical practice. In this respect, DL-based methods contribute more to RPT dosimetry workflows.

The pivotal subject of the included radiomics studies was RLT and PRRT response prediction. Several studies extracted RFs from radioisotopes labeled-SSTR and PSMA PET/CT to predict treatment response/outcomes. Even though most of the literature is focused on [^68^Ga]Ga-SSTR and PSMA PET/CT, there are a few studies on [^64^Cu]Cu-DOTA-TATE [109,110] and [^18^F]PSMA-1007 [97].

Since conventional PET parameters, such as SUV and volume, showed controversial results regarding dose prediction or PRRT response or outcomes [66,72], several studies employed texture features and obtained promising results for response prediction. Interestingly, Laudicellaet et al.’s [72] and Önner et al.’s [67] results in PRRT response prediction were similar in the best-obtained RFs, skewness, and kurtosis.

Employing [^68^Ga]Ga-SSTR and PSMA PET/MRI RFs for RPT purposes is scarce in the literature, and the results are controversial. Weber et al. [68] utilized TFs from [^68^Ga]Ga-SSTR PET/MRI and obtained no parameters from either PET or ADC maps that could predict PRRT response [68]. However, the results of extracting RFs from [^68^Ga]Ga-PSMA PET/MRI by Roll et al. [49] are promising.

In the case of extracting RFs from [^68^Ga]Ga-PSMA PET, some works highlighted the importance of employing ML models to predict RLT outcomes/responses [41,46,47,48,49]. The implementation of RFs from [68Ga]Ga-SSTR and PSMA PET/CT provided profound prognostic information regarding the restaging of patients with high-risk NETs or mCRPC through different studies. In some cases, ML algorithms were employed to differentiate benign from malignant [54], to classify M- and N-stages [56], or to discriminate bone metastases [57] with significant results.

Some studies focused on implementing RFs from radiolabeled-SSTR and PSMA PET/CT for semi-automatic segmentation. The authors used different methods, such as threshold-based [60,100], edge-based [73], and gradient edge [74], to increase the delineated targets’ robustness. Some authors proposed practical solutions to quantify [^68^Ga]Ga-PSMA and [^64^Cu]Cu-DOTA-TATE PET/CT biomarkers semi-automatically [58,59,60,109,110].

Several studies have been dedicated to predicting ADs before PRRT and RLT using extracted RFs from pre-therapeutic [^68^Ga]Ga-SSTR and PSMA PET/CT scans. This idea was first proposed in 2012 to find a correlation between conventional parameters of pre-therapy [^68^Ga]Ga-DOTA-TOC PET scan with the dose distribution of post-RPT and has continued until now [78]. As a preliminary study, Xue et al. [63,64] recently used ML techniques with RFs extracted from [^68^Ga]Ga-PSMA PET/CT imaging and proved the feasibility of pre-therapeutic RLT dosimetry estimation.

To date, the scientific literature lacks studies that specifically address the pre-therapy dose prediction of ^90^Y and ^225^Ac RPTs. Furthermore, in the context of ^177^Lu RPT, no study employed other radioisotopes-labeled-SSTR and PSMA than ^68^Ga for this task.

Finally, a notable advancement in RLT and PRRT is the integration of DL techniques into the dosimetry workflow. Image-based dosimetry workflow from image acquisition to dose estimation is very complicated, takes significant time and effort, and is prone to human errors. Accordingly, DL-based methods have been introduced to alleviate dosimetry complexity in each step. In [^177^Lu]Lu-SSTR–or PSMA RPT dosimetry, a few studies implemented DL methods in image quantification [128], image segmentation [134,135,136], and dose estimation [139,140,141,143]. Furthermore, Dewaraja et al. [158] developed fully automated voxel dosimetry in PRRT, including automated organ segmentation, registration, dose-rate estimation, and curve-fitting steps in the internal dosimetry workflow.

There is missing research in the literature, to our knowledge, on the use of AI-based methods in the dosimetry workflow at ^225^Ac and ^90^Y RPT. Considering the quantitative imaging challenges associated with ^225^Ac and ^90^Y RPT, AI emerges as a promising solution to address this issue, thereby presenting a potential avenue for future research [148,149,150,151]. Despite the significant and promising results of the studies that employed DL in the dosimetry workflow, the literature body is inadequate, especially in the case of ^90^Y and ^225^Ac. Further investigation is required, emphasizing clinical applications.

Different studies showed that RFs extracted from radiolabeled-SSTR and PSMA PET scans can play an essential role in various tasks. However, there were several variations, such as variability in study design, scanner setup, patient selection, sample size, patient movement, injected activity, segmentation methods, RFs computation, image reconstructive procedures, and data analysis methodologies.

In terms of ML analysis, the variability in the distribution of the dataset, training and validation methods, optimization process, and the choice of hyperparameters and ML algorithms has an enormous impact on the interpretation of the final results. Furthermore, the presence of variabilities significantly hampers the robustness, reproducibility, and repeatability of the extracted RFs. Consequently, the current feasibility of applying RFs derived from radiolabeled-SSTR and PSMA PET/CT or PET/MRI in clinical PRRT or RLT practice remains limited.

Non-Gallium-68 radiopharmaceuticals, such as [^18^F]PSMA-1007, [^64^Cu]Cu-PSMA-617, [^64^Cu]Cu-DOTA-TATE, and [^44^Sc]Sc-PSMA-617, with a longer half-life than gallium-68 and ability to detect small lesions, can potentially be used further for different tasks, such as RPT response monitoring and pre-therapeutic dose prediction, to improve accuracy and speed.

Radiomics has shown its capability for response prediction, restaging, segmentation, and dose prediction in theranostics radiolabeled-SSTR and PSMA pairs. Therefore, in the future, more radiomics studies should be conducted with large sample sizes. Moreover, the robustness, standardization, harmonization, repeatability, and reproducibility of RFs should be considered. In this sense, independent datasets for model validation are a practical solution. For more valuable results in the future, combining imaging RFs with clinical features and imaging phenotypes would be essential.

Another issue to consider in future research is data sharing for more investigations into radiomics’ usefulness in improving results close to routine clinical practice. Moreover, developing more AI methods for dosimetry estimation and prediction is recommended. There are measures computed from 3D absorbed dose maps called dosiomic features that can be used to train AI models in outcome prediction and enable further personalized therapies. DL approaches possess the inherent capability to identify reliable predictors of outcome and can aid in the development of models for tumor control probability and radiation-induced normal tissue toxicity.

Furthermore, augmenting AI models with additional information, such as radiobiological parameters and physical equations, holds the potential to yield more reliable and accurate results. By incorporating these factors into the AI modeling process, we can enhance the understanding of the underlying radiobiology and physical interactions involved in PRRT and RLT. This integrated approach has the potential to improve treatment planning, optimize dose predictions, and ultimately enhance the overall efficacy and safety of PRRT and RLT procedures.

In precision medicine, digital twin models as virtual representations of a patient use real-time data along with simulation and ML to help decision-making [159]. Similarly, in precision oncology, by using digital twins, a disease and its suitable treatments can be accurately modeled [160]. In this regard, Rahmim et al. [161] envisioned theranostic digital twins (TDTs) as a way of overcoming one-size-fits-all therapeutic schemes in the future. TDTs can efficiently and accurately design personalized RPTs, including optimized intervention parameters. Examples are optimizing injected radioactivities, injection sites, injection intervals and profiles, and combining therapies. As a result of the use of all available patient-specific information, such as multimodal, multiscale images, combined with other data and assisted using AI techniques, it is possible to simulate different treatment scenarios. Patients are often digitally twinned in this manner, allowing clinicians to provide a higher level of care and select more effective RPTs (e.g., individualized injected radioactivity). In addition, TDTs can be continuously updated with new patient data (see Figure 2 in reference [161]) to ensure personalized treatment toward more optimal outcomes [161,162,163,164].

## 11. Conclusions

We have provided an elaborate overview of state-of-the-art radiomics and AI applications for theranostic applications of radioligands targeting SSTR and PSMA. Such applications have the significant potential to improve diagnosis and treatment monitoring by adding quantitative information to an expert’s visual analysis. Considering the promising results in the presented studies, it will be necessary for future researchers to reproduce and validate these findings with a sufficient population of patients. This will pave the way for generating significant scientific evidence to translate potential applications of radiomics and AI into clinical practice. It will also expand the use of dosimetry-based therapies in managing patients undergoing RPTs. To overcome the difficulties in the reproducibility and generalization of radiomics and AI studies, conducting multi-center studies through data sharing and harmonization seems indispensable. It is important to focus on the use of radiomic signatures, biological properties, and physical models in network architecture training. Overall, such applications have the tremendous potential to further enhance patient outcomes.

## Figures and Tables

**Figure 1 diagnostics-14-00181-f001:**
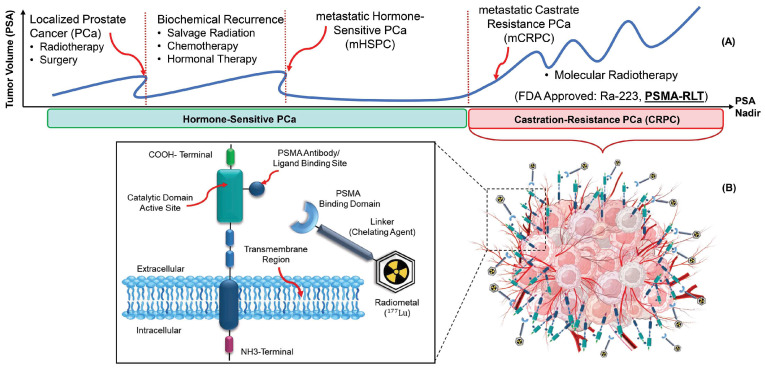
The principle of radiotheranostics in mCRPC patients. (**A**). The typical timeline of different therapies, including RPT (also known as RLT). (**B**). PSMA-binding domain, linker, and chelator labeled with Lu-177 deliver ionizing radiation to the tumor.

**Figure 2 diagnostics-14-00181-f002:**
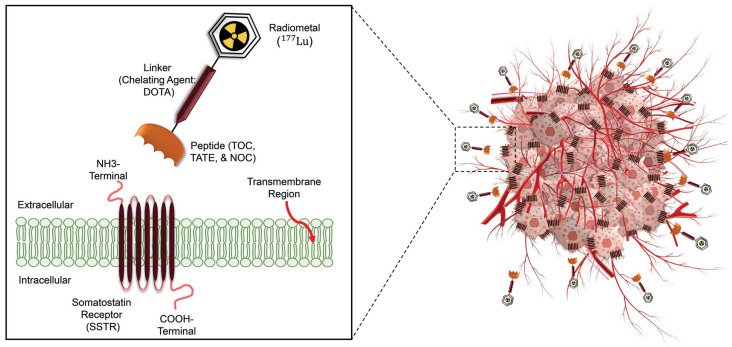
Schematic overview of radiotheranostics principle in NETs patients and the development of radiopharmaceutical. The chelator labeled with Lu-177 binds to SSTRs and delivers ionizing radiations to destroy tumor cells.

**Figure 3 diagnostics-14-00181-f003:**
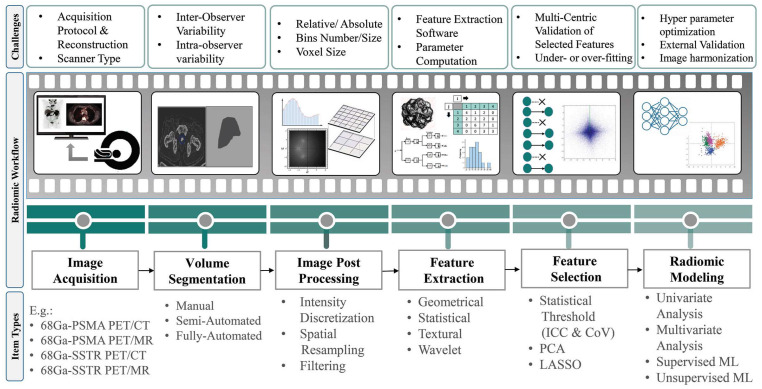
Radiomics and AI workflow from image acquisition to radiomics modeling.

**Figure 4 diagnostics-14-00181-f004:**
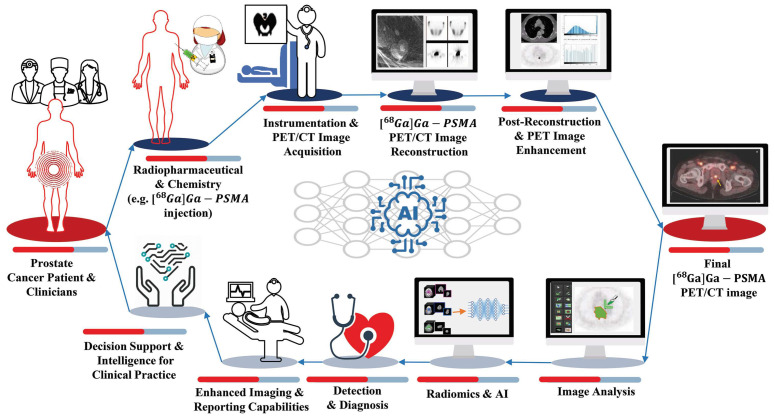
A broad range of AI applications in a chain from radiochemistry to a physician’s report generation for a prostate cancer patient who underwent [^68^Ga]Ga-PSMA-11 PET/CT scan based on SNMMI AI task-force guideline.

**Figure 5 diagnostics-14-00181-f005:**
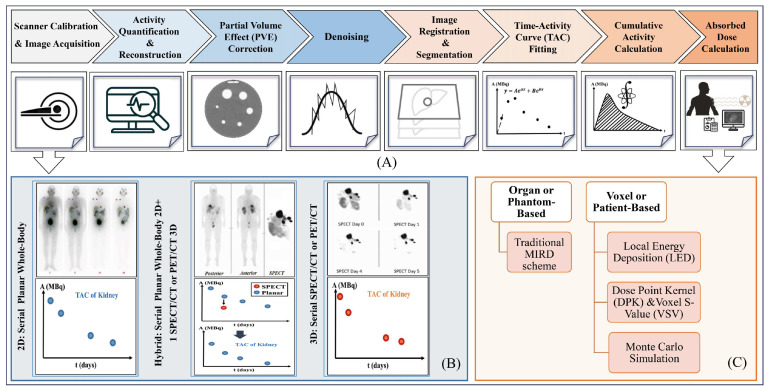
(**A**). Internal dosimetry workflow for dose calculation. (**B**). Three acquisition protocols to provide TAC for delineated targets on serial images. (**C**). Organ- and voxel-based dose calculation methods with their subsets.

**Table 1 diagnostics-14-00181-t001:** Radiotheranostic pairs and targets in NETs and mCRPC diseases with an emphasis on clinical relevance.

Therapeutic Radioisotopes	Diagnostic Radioisotopes-Pharmaceuticals	
	SSTRs Target/NET	PSMA Target/mCRPC
^177^Lu	[^68^Ga]Ga-DOTA-TATE PET	[^68^Ga]Ga-PSMA-617 PET
	[^68^Ga]Ga-DOTA-TOC PET	[^68^Ga]Ga-PSMA-I&T PET
		[^68^Ga]Ga-PSMA-11 PET
	[^64^Cu]Cu-DOTA-TATE PET	[^64^Cu]Cu-PSMA-617 PET
	[^64^Cu]Cu-DOTA-TOC PET	
	No Clinical Match	[^18^F]PSMA-617 PET
	No Clinical Match	[^44^Sc]Sc-PSMA-617 PET
^225^Ac	[^177^Lu]Lu-DOTA-TATE SPECT	[^177^Lu]Lu-PSMA-617 SPECT
	[^177^Lu]Lu-DOTA-TOC SPECT	
^90^Y	[^177^Lu]Lu-DOTA-TATE SPECT	[^177^Lu]Lu-PSMA-617 SPECT
	[^177^Lu]Lu-DOTA-TOC SPECT	[^177^Lu]Lu-J591 SPECT
	[^111^In]In-DOTA-TATE SPECT	[^111^In]In-J591 SPECT
	[^111^In]In-DOTA-TOC SPECT	

These clinical radiotheranostics pairs are listed for completeness, but they are not discussed further.

**Table 2 diagnostics-14-00181-t002:** Summary of radiomics and AI studies on [^68^Ga]Ga-PSMA-11/[^177^Lu]Lu -PSMA-617 radiotheranostic pairs.

#	First Author, Year [Ref]	Radiopharmaceutical, Modality	# Pats	Site	Utility	Feature Class	Stats, ML/DL Algorithms	Software	Finding RFs	Result	Conclusion
1	Grubmüller et al., 2018 [39]	[^68^Ga]Ga-PSMA-11 PET/CT	38	77 primary prostate and metastatic LNs, bone and visceral metastases	OS prediction	First order (shape and intensity)	Unavailable Cox proportional hazards model, KM, and Cohen’s kappa (κ)	Hermes Hybrid3D (Hermes Medical Solutions, Stockholm)	TTV	TTV was significantly associated with OS, and its changes were significantly associated with PSA response (*p* = 0.58), contrary to SUV_mean_ changes (*p* = 0.15)	PSMA-TTV is a promising tool for RPT response evaluation
2	Khurshid et al., 2018 [40]	[^68^Ga]Ga-PSMA-11 PET/CT	70	118 primary prostate and metastatic LNs, bone and liver metastases	Therapy response prediction	First order (intensity)/second order (texture)	Spearman correlation	NM	NGLCM (Entropy and homogeneity)	Entropy (r = -0.327) and homogeneity (r = 0.315) TFs of bone lesions correlated with ∆PSA	Better treatment response for more heterogeneous lesions
3	Acar et al., 2019 [41]	[^68^Ga]Ga-PSMA-11 PET/CT	75	257 metastatic bone lesions	Therapy response prediction	First order (shape and intensity)/second order (texture)	Decision tree, discriminant analysis, SVM, KNN, and ensemble classifier	LIFEx	GLZLM_SZHGE and histogram-based kurtosis	Weighted KNN achieved the best classification performance with AUC = 0.76 (ACU = 73.5%, SE = 73.5%, SP = 73.7%)	Metastatic or responded sclerotic bone lesions discrimination using CT texture analysis and ML
4	Seifert et al., 2020 [42]	[^68^Ga]Ga-PSMA-11 PET/CT	110	136 metastatic LNs, bone and visceral (liver, lung, and pleura) lesions	OS prediction/restaging/Seg	First order (shape and intensity)/	Univariate and multivariate regression, Spearman correlation, and Mann–Whitney U tests	MIWBAS, V.1.0, Siemens	PSMA-TV	Lesion number (HR = 1.255), PSMA-TV (HR =1.299), and PSMA-TLQ (HR = 1.326) prognosticators of OS	- Baseline PSMA-PET TV was a significant negative prognosticator of OS in prostate cancer before RPT - In comparison with PSMA-TV, PSMA-TLQ was an independent and superior prognosticator of OS
5	Widjaja et al., 2021 [43]	[^68^Ga]Ga-PSMA-11 PET/CT	71	208 primary prostate and metastatic LNs, bone, liver, and soft tissue lesions	Biochemical response prediction	First order (shape and intensity)	Kruskal–Wallis, Fisher’s exact, and KM	syngo.via; V50B; Siemens	SUV_max_	SUV_max_ was an independent predictor for early PSA response in the treatment course	Higher PSMA expression was related to a better early biochemical response
6	Gafita et al., 2021 [44]	[^68^Ga]Ga-PSMA-11 PET/CT	414	463 metastatic LNs, bone, and liver lesions	OS and PFS prediction	First order (intensity)	LASSO, Wilcoxon, and Mann–Whitney	qPSMA V.1.0	SUV_mean_	PSM SUV: correlated significantly with tumor PSMA expression	- Higher PSMA expression correlated with longer OS and PSA-PFS - Patients with metastatic bone disease had shorter OS and PSA-PFS
7	Khreish et al., 2021 [45]	[^68^Ga]Ga-PSMA-11 PET/CT	51	322 primary prostate and metastatic LNs, bone, liver, and soft tissue lesions	PFS prediction	First order (intensity)	KM, Cox proportional-hazards modeling, Spearman, and Cohen’s κ	NM	TLR	ΔTLR and ΔSUV significantly correlated with ΔPSA. Univariate analysis: SUV_peak_ failed to predict survival Multivariable analysis: TLR was independently associated with PFS	TLR (normalization of the total lesion PSMA over healthy liver tissue uptake) biomarker can be a predictor of PFS in RPT
8	Moazemi et al., 2021 [46]	[^68^Ga]Ga-PSMA-11 PET/CT	83	2070 primary prostate and metastatic lesions	Therapy response prediction	First order (intensity)/second order (texture)	5 ML classifiers (linear, RBF, and polynomial kernel SVMs), ET, and random forest)	InterView Fusion (V. 3.08.005)	Task I: PET (Min and Correlation) and CT (Min, Coarseness, and Busyness)	Strong correlations between ML SVM classifier (RBF kernel) on a selection of RFs and clinical parameters with ΔPSA (with AUC = 80%, SE = 75%, and SP = 75%)	RFs were superior to clinical parameters in terms of correlation with ΔPSA
9	Moazemi et al., 2021 [47]	[^68^Ga]Ga-PSMA-11 PET/CT	100	2067 pathological hotspots	Therapy response prediction/auto Seg	First order (shape and intensity)/second order (texture)	UNet and 6 ML classifiers (logistic regression, SVM (linear, polynomial, RBF kernels), ET, and random forest)	PyRadiomics Library	14 features from both PET and CT modalities	Seg. task (0.88 precision, 0.77 recall, and 0.82 Dice). In predicting the response task, logistic regression performed the best (with AUC = 0.73, SE = 0.81, and SP = 0.58)	In ^177^Lu-PSMA RPT, the facilitated automated decision support tool has an assistant potential for patient screening
10	Moazemi et al., 2021 [48]	[^68^Ga]Ga-PSMA-11 PET/CT	83	2070 primary prostate and metastatic lesions	OS prediction/restaging	First order (shape and intensity)/second order (texture)	LASSO regression and KM estimator	InterView Fusion (V. 3.08.005)	PET kurtosis and SUV_min_	The relevant RFs significantly correlated with OS (r = 0.2765, *p* = 0.0114)	^68^Ga-PSMA-PET/CT scans and patient-specific clinical parameters have the potential for the prediction of OS in advanced PC patients under ^177^Lu-PSMA RPT
11	Roll et al., 2021 [49]	[^68^Ga]Ga-PSMA-11 PET/MRI	21	49 metastatic lesions in bone, LNs, liver, and lung	Biochemical response and OS prediction	First order (intensity)	KM analysis and log-rank test	3D slicer, V.4.11.2	T_2_-weighted (interquartile range)	The logistic regression model revealed the highest accuracy (AUC = 0.83)	There was a high survival for patients with a biochemical response and higher T_2_ interquartile range values
12	Rosar et al., 2022 [50]	[^68^Ga]Ga-PSMA-11 PET/CT	66	139 metastatic lesions in bone, LNs, liver, and other soft tissue	OS prediction	First order (shape and intensity)	Spearman’s rank correlation and KM	Syngo. Via (Enterprise VB 40B, Siemens, Erlangen, Germany)	TLP	There was a strong correlation between ∆PSA and ∆TLP (r = 0.702)	TLP (summed products of volume × uptake (SUV_mean_) of all lesions) biomarker independently and strongly predicted OS
13	Gafita et al., 2022 [51]	[^68^Ga]Ga-PSMA-11 PET/CT	406	normal liver, spleen, salivary gland and kidney, and metastatic lesions in bone, LNs, and visceral organs	Therapy response prediction/restaging	First order (shape and intensity)	Spearman CC and Kruskal–Wallis testing	qPSMA	PSMA-VOL	Salivary glands, kidneys, and liver: a moderate and negative correlation between PSMA-VOL and SUV_mean_ Spleen: a weak correlation between PSMA-VOL and SUV_mean_	Decreasing the activity concentration in OARs due to the tumor sequestration affecting the biodistribution of ^68^Ga-PSM showed the tumor sink effect
14	Hartrampf et al., 2022 [52]	[^68^Ga]Ga-PSMA-11 PET/CT	65	144 primary prostate and metastatic bone, LNs, liver, and lung lesion	Therapy response assessment	First order (shape and intensity)	Shapiro–Wilk tests and Spearman’s rank CC	FIJI (ImageJ)	ΔPSMA-TV	ΔPSA was correlated with ΔSUV_maxall_ (r = 0.51), ΔPSMA-TV_all_ (r ≥ 0.59), ΔPSMA-TV_10_ (r ≥ 0.57), and ΔPSMA-TV_5_ (r ≥ 0.53)	The RPT response assessment was possible through PSMA-TV
15	Pathmanandav et al., 2022 [53]	[^68^Ga]Ga-PSMA-11 PET/CT/[^18^F]FDG PET/CT	56	92 metastatic lesions in bone, LNs, and visceral organs	Therapy response prediction	First order (shape and intensity)	KM, Cox proportional-hazards regression, logistic regression, and LASSO	MIM	PSMA_TV and SUV_mean_	PSMA SUV_mean_ was an independent predictor of treatment response, but SUV_max_ was not	A higher SUV_mean_ correlated with treatment response, but a higher PSMA_TV was associated with worse OS
16	Giesel et al., 2017 [54]	[^18^F]FDG PET/CT, [^68^Ga]Ga-PSMA-11 PET/CT, and [^68^Ga]Ga-DOTA-TOC PET/CT	148 (40 PCa)	254 metastatic LNs	Restaging	First order (shape and intensity)	2-sided paired-sample *t*-test, 2-sided Wilcoxon signed-rank testing	In-house (developed at the Fraunhofer Institute for Medical Image Computing)	PET (SUV_max_) CT (short-axis diameter (SAD) and Histogram)	CT densities correlated with the PET uptake (with a 7.5 HU threshold to discriminate between malignant and benign LNs infiltration) and 20 HU to exclude benign LN	CT density measurements and PET uptake analysis increased the differentiation between malignant and benign LN
17	Moazemi et al., 2020 [55]	[^68^Ga]Ga-PSMA-11 PET/CT	72	2419 hotspots in normal kidney, bladder and salivary glands, and metastatic lesions	Restaging	First order (shape and intensity)/second order (texture)	5 ML classifiers (SVM (linear, RBF, and polynomial kernels), ET, and random forest)	InterView FUSION (V3.08.005)	PET (kurtosis; busyness, and coarseness)	AUC = 0.98, (SE = 0.94 and SP = 0.89) ET and RF showed the best results	Using ML and considering features from both the CT and PET images outperformed using either separately
18	Erle et al., 2021 [56]	[^68^Ga]Ga-PSMA-11 PET/CT	87	2452 hotspots in normal liver, kidney, lacrimal and salivary glands, and metastatic lesions	Restaging	First order (intensity)/second order (texture)	SVM (linear kernel), ET, and random forest	InterView FUSION	77 RFs	The ET classifier resulted in (AUC = 0.95, SE = 0.95, and SP = 0.80)	Combining manual and ML-based diagnoses has the potential to predict hotspot labels with high sensitivity
19	Hinzpeter et al., 2021 [57]	[^68^Ga]Ga-PSMA-11 PET/CT	67	205 bone metastases	Restaging	First order (intensity)/ second order (texture)	Gradient-boosted tree	3D Slicer, V.4.11	11 most important and independent features^2^	Model classification AUC = 0.85 (with SE = 78% and SP = 93%)	The distinction of healthy bone from metastatic bone accurately using PET/CT texture analysis and ML
20	Hammes et al., 2018 [58]	[^68^Ga]Ga-PSMA-11 PET/CT	38	100 metastatic bone lesions	Staging/therapy response prediction/Seg	First order (intensity)	Linear regression and ANOVA	NA	SUV_max_ and SUV_mean_	SUV_max_, r^2^ = 0.97; SUV_mean_, r^2^= 0.88; lesion count, r^2^ = 0.97; HU threshold: not significant	EBONI has the potential to semi-automatically quantify TVs in PSMA PET/CT in a fast (3 min per scan), robust, and reproducible manner
21	Zhao et al., 2019 [59]	[^68^Ga]Ga-PSMA-11 PET/CT	193	1756 primary prostate and metastatic lesions in bone and LNs	Staging/restaging/Seg	NA	2.5DU-Net	NA	NA	Bone lesion detection (precision = 99%, recall = 99%, and F1 score = 99%), LN lesion detection (precision = 94%, recall = 89%, and F1 score = 92%)	CNN has the potential to automatically segment disease sites on ^68^Ga-PSMA PET/CT images to confirm whether a voxel is a lesion or not
22	Seifert et al., 2020 [60]	[^68^Ga]Ga-PSMA-11 PET/CT	40	100 metastatic lesions in the bone, LNs, liver, and lung	Seg/OS prediction	First order (shape and intensity)	Seg: GAN *t*-tests, log-rank tests, Cox regression, ICC, Pearson correlation	MIWBAS, V.1.0	PET_TV_50_	PSMA_TV50_: R^2^ = 1.000 and SUV_max_: R^2^ = 0.988	PSMA_TV50_ was a significant predictor of OS
23	Xue et al., 2020 [61]	^[68^Ga]Ga-PSMA-11 PET/CT	30	Main organs and tumor lesions	Dose prediction	NA	GAN	NA	NA	The dual-input-model is able to synthesize dose maps with MAPE of 18.94% ± 5.65%	AI is capable of estimating voxel-wise posttherapy dosimetry both qualitatively and quantitatively
24	Xue et al., 2021 [62]	^[68^Ga]Ga-PSMA-11 PET/CT	34	Main organs and tumor lesions	Dose Prediction	NA	GAN	NA	NA	DVH: MAE = 21.2 ± 10.8% (=24.0 ± 10.0% without pre-training) to the ground truth	Using the PBRK model along with a pre-therapeutic PET/CT image may improve the development of AI for dose prediction
25	Xue et al., 2022 [63,64]	[^68^Ga]Ga-PSMA-11 PET/CT	23	WB, kidney, liver, spleen, and salivary gland	Dose prediction	First order (shape and intensity)	RFR and ANN	NA	SUV_max_ and TV	The dose prediction based on the literature population means had a significantly larger MAPE for each organ compared to the optimal ML methods Average prediction error for kidneys = 15.76%	It is possible to estimate the dose before RPT, which may support the treatment planning role

ACU: accuracy; HR: hazard ratio; NA: not applicable; NM: not mentioned; RBF: radial basis function; Resp: respectively; SE: sensitivity; Seg: segmentation; SP: specificity.

**Table 3 diagnostics-14-00181-t003:** Summary of radiomics studies on ^68^Ga-SSTR/^177^Lu-SSTR radiotheranostic pairs.

#	First Author, Year [Ref]	Radiopharmaceutical, Modality	# pats	Site	Utility	Feature Class	Stats, ML/DL Algorithms	Software	Finding RFs	Result	Conclusion
1	Werner et al., 2017 [65]	[^68^Ga]Ga-DOTA-TATE PET/CT	142	872 primary tumors of GEP-NETs (pancreatic, stomach, and intestine), lung and metastatic lesions in LNs, bone, liver, and lung	OS and PFS prediction	First order (intensity)/second order (texture)	Cox multi-parametric regression, Youden index, and KM	Interview FUSION	Entropy, correlation, short zone emphasis and homogeneity	Eight statistically independent TFs for time-to-progression and time-to-death were identified with Cox analysis, among which it was entropy that predicted both PFS and OS	The prognostic performance of intratumoral TFs analysis outperformed conventional PET parameters
2	Werner et al., 2018 [66]	[^68^Ga]Ga-DOTA-TATE/DOTA-TOC PET/CT	31	162 metastatic lesions in LNs, bone, liver, and lung	OS prediction	First order (intensity)/second order (texture)	Youden index, KM, multivariate Cox hazard analysis, and relative risks	Interview Fusion	Entropy	SUV_mean/max_ was not able to prognosticate Entropy was a significant RF to distinct high- and low-risk groups	Unlike conventional PET parameters, higher entropy (a texture feature) values were associated with more prolonged survival
3	Önner et al., 2020 [67]	[^68^Ga]Ga-DOTA-TATE PET/CT	22	326 primary tumors of the pancreas, stomach, intestine, and metastatic lesions in LNs, bone, liver, and lung	Treatment response prediction	First order (intensity)/second order (texture)	Kolmogorov–Smirnov, Mann–Whitney U, and Youden index	LIFEx	Skewness and kurtosis	AUC: for skewness and kurtosis (0.619 and 0.518, resp.)	Skewness and kurtosis predicted PRRT response
4	Weber et al., 2020 [68]	[^68^Ga]Ga-DOTA-TOC PET/MRI	9 PRRT	80 metastatic liver lesions	Treatment response prediction	First order (intensity)/second order (texture)	Mann–Whitney test	LIFEx	ADC maps (lesion vol and entropy)	No PET parameter values predicted PRRT response In the treatment responders group: a significant decrease in ADCmaps_lesion volumes and ADCmaps_entropy	No parameters of PET or ADC maps predicted PRRT response. However, the study sample size was small, so further research is suggested
5	Ortega et al., 2021 [69]	[^68^Ga]Ga-DOTA-TATE PET/CT	91	872 primary tumors of GEP-NETs (pancreatic, intestine, and stomach), lung and metastatic lesions in LNs, bone, liver, and lung	PFS prediction	First order (intensity)/second order (texture)	2-sided Wilcoxon rank sum test and Cox proportional hazards model	PACS system with fusion software (Mirada Medical)	Multivariate analysis: mean SUV_max_ and mean lesion SUV_max_/liver SUV_max_	Significantly higher mean SUV_max_ in responders than that in non-responders A higher mean SUV_max_ and mean SUV_max_ tumor-to-liver ratio was associated with therapy response - Higher kurtosis values were observed in non-responders than in responders (mean 8.6 vs. 5.8)	SSTR expression and tumor heterogeneity metrics associated with PFS
6	Atkinson et al., 2021 [70]	[^68^Ga]Ga-DOTA-TATE PET/CT	44	GEP-NETs primary tumors (pancreatic, stomach, intestine), lung, thyroid and phaeochromocytoma/paraganglioma and metastatic lesions in LNs, bone, liver, lung, peritoneum, and brain	OS and PFS prediction	First order (intensity)/second order (texture)	Univariate KM and multivariate Cox regression	TexRAD, Cambridge, UK	CT-coarse kurtosis, PET_entropy, and PET_skewness	SUV_max_ and SUV_mean_ were not significant in outcome prediction Higher kurtosis, higher entropy, and lower skewness: predict shorter PFS CT-TA (coarse kurtosis): independently predicates PFS (HR = 2.57 and CI = 1.22–5.38) PET-TA (unfiltered skewness): independently predicates OS (HR = 9.05, 95% CI = 1.19–68.91)	Texture analysis yielded prognostic biomarkers that had the potential to assess outcomes in NETs patients with more aggressive diseases
7	Liberini et al., 2021 [71]	[^68^Ga]Ga-DOTA-TATE PET/CT and [18F]FDG PET/CT	2	22 metastatic lesions in LNs, bone, and liver	Prognosis prediction	First order (intensity)/second order (texture)	Mann–Whitney, Pearson correlation matrix, and PCA	LIFE xV.5.10 (IMIV/CEA, Orsay, France)	TLSRE_wb-50_ and SRETV_wb-50_	Mann–Whitney test: 28 RFs showed significant differences between the two patients Pearson correlation matrix: identified seven second-order RFs, with poor correlation with SUV_max_ and PET vol.	Defining inter-patient heterogeneity and therapy response prediction may be possible using RFs
8	Laudicella et al., 2022 [72]	[^68^Ga]Ga-DOTA-TOC PET/CT	38	324 metastatic lesions in LNs, bone, liver, and other soft tissues	Treatment response prediction	First order (intensity)/second order (texture)	*t*-test, Mann– Whitney U, and Youden index	LIFEx	HISTO_Skewness and HISTO_Kurtosis	HISTO_Skewness and HISTO_Kurtosis: able to predict the response (AUC ROC, SE., and SP. of 0.745, 80.6%, 67.2% and 0.722, 61.2%, 75.9%, resp.) SUV_max_ was not able to predict the response (AUC= 0.523)	The developed theragnomics (THERAGNOstics + radiOMICS) predictive model was superior to conventional quantitative parameters to predict the GEP-NET lesion’s response to ^177^Lu-DOTA-TOC PRRT
9	Giesel et al., 2017 [54]	[^18^F]FDG PET/CT, [^68^Ga]Ga-PSMA-11 PET/CT, and [^68^Ga]Ga-DOTA-TOC PET/CT	148 (35 GEP-NET)	217 metastatic LNs	Restaging	First order (shape and intensity)	2-sided paired-sample t-testing, 2-sided Wilcoxon signed-rank testing	In-house (developed at the Fraunhofer Institute for Medical Image Computing)	PET (SUV_max_), CT (short-axis diameter (SAD) and histogram)	CT densities correlated with the PET uptake (with a 7.5 HU threshold to discriminate between malignant and benign LNs infiltration and 20 HU to exclude benign LN)	CT density measurements and PET uptake analysis increased the differentiation between malignant and benign LNs
10	Liberini et al., 2021 [73]	[^68^Ga]Ga-DOTA-TOC PET/CT	49	60 primary tumors of GEP-NETs (pancreatic, stomach, intestine) and metastatic lesions in LNs, liver, and other soft tissues	Prognosis prediction/seg./restaging	First order (intensity)/second order (texture)	Pearson’s CCs, DSC, ICC, and coefficient of variance	LifeX V.4.81 (IMIV/CEA, Orsay, France)	GLZLM (also called GLSZM) features and zones with low gray-level (SZLGE and LZLGE), and SUV_max_ thresh. of 40%	SAEB seg. and operators: DSC mean= 0.75 ± 0.11 (0.45–0.92), SAEB seg. and 4 manual segs.= 0.78 ± 0.09 (0.36–0.97)	Superior RFs stability among operators was provided using SUV_max_ thresholds of 40% but led to a possible biological information loss SAEB performed better than manual segmentation; however, further validation is suggested
11	Wehrend et al., 2021 [74]	[^68^Ga]Ga-DOTA-TATE PET/CT	125	223 liver lesions	Seg	NA	CNN: 2D U-Net Stats: F1 score	MIM (V. 7.03)	NA	Highest precision-recall AUC (0.73 ± 0.03): using a noise filter (15-pixel) Highest mean PPV (0.94 ± 0.01): 20-pixel filter Highest mean F1 score (0.79 ± 0.01): 20-pixel filter Highest mean SE. (0.74 ± 0.02): 5-pixel filter	DNN can automatically facilitate the detection of hepatic metastases For further validation, it suggested the need for more studies with larger sample sizes
12	Akhavanallaf et al., 2023 [75]	[^68^Ga]Ga-DOTA-TATE PET/CT	25	90 NETs: 75 liver, 11 LNs, 3 primary pancreas tumors, and 1 chest tumor	Dose prediction	First order (shape and intensity)	Spearman rank correlation, univariate linear regression model, ElasticNet and permutation-based RF variable-importance feature selection	NM	SUV_mean_, TLSUV_mean_ (SUV_mean_ of total-lesion-burden), and SUV_peak_	Tumor dose prediction using an optimal trivariate RF model composed of SUVmean, TLSUVmean, and total liver SUVmean: R2 = 0.64, MAE = 0.73 Gy/GBq, and MRAE = 0.20	PET-based metrics combined with ML models can improve dose prediction, which may be useful for stratifying patients and personalizing treatment
13	Plachouris et al., 2023 [76]	[68Ga]Ga-DOTA-TOC PET/CT	20	3412 features from 4 OARs (liver, spleen, and left- and right kidneys)	Dose prediction	First order (intensity)/second order (texture) + dosiomic features	Multivariate analysis and nine supervised linear and non-linear-based ML regression algorithms: linear, ridge, extra tree, AdaBoost, gradient boost, random forest, decision tree, SVR, and XGBoost regression algorithms trained for every OAR	PyRadiomics Library	Differed for each OAR (Table 3 in [76])	Wavelet-based features had highly correlated predictive value More precise prediction using non-linear-based ML regression algorithms than linear-based ones	The combination of radiomics and dosiomics may be useful for individualized molecular radiotherapy response assessment and OAR dose prediction

ACU: accuracy, Stats: statistics; ANOVA = analysis of variance; DSC: dice similarity coefficient; NA: not applicable; NM = not mentioned; Seg: segmentation; Resp: respectively.

## Data Availability

Not applicable.

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
