# Peer review of "Radiomics and Artificial Intelligence in Radiotheranostics: A Review of Applications for Radioligands Targeting Somatostatin Receptors and Prostate-Specific Membrane Antigens"

_diagnostics, 2024, doi:10.3390/diagnostics14020181_

Round 1

Reviewer 1 Report

Comments and Suggestions for Authors

Dear authors,

Thank you for this elaborate review of the latest radiomics and AI methods for pre-therapeutic radiolabeled-SSTR and PSMA PET/CT or PET/MR images for different tasks, such as treatment response prediction, restaging, segmentation, and dosimetry using various analysis methods, including univariate and multivariate statistical analysis and ML approaches.

Covering the usefulness of -omics for these treatment techniques is crucial nowadays, as those are quickly developing and important therapeutical options.

From the image analysis point of view, it seemed like the review lacked references cinetic 3D models for pre-treatment dosimetry.

From the pharmaceutical point of view, the Table 1 lists all the RPTs that exist, but it is not clear enough :  what is the inerest in white lines in the diagnostic part?The  fact that only the radio-nuclei are mentionnes and not the pharmaceutics associated, does it mean that all the RPs cited in the diagnostic radio-isotopes pharmaceuticals part were also tested with the therapeutical isotope referred to in the table? Does it mean that for AC225 that was no tests done with Gallium as a theranostic couple? If it is that, there are studies that compared the two in pre-clinical studies : Meyer C, Prasad V, Stuparu A, Kletting P, Glatting G, Miksch J, Solbach C, Lueckerath K, Nyiranshuti L, Zhu S, Czernin J, Beer AJ, Slavik R, Calais J, Dahlbom M. Comparison of PSMA-TO-1 and PSMA-617 labeled with gallium-68, lutetium-177 and actinium-225. EJNMMI Res. 2022 Oct 1;12(1):65. doi: 10.1186/s13550-022-00935-6. PMID: 36182983; PMCID: PMC9526774.

Theranostic Nuclear Medicine with Gallium-68, Lutetium-177, Copper-64/67, Actinium-225, and Lead-212/203 Radionuclides Focus Review
Katherine A. Morgan, Stacey E. Rudd, Asif Noor, and Paul S. Donnelly
Chemical Reviews 2023 123 (20), 12004-12035
DOI: 10.1021/acs.chemrev.3c00456

Then I do not understand what this table is clarifying compared to the list mentionned in the text.

And small erros :

Table 1, column SSTRs Target/NET, 3rd line : ']' is misplaced.It should be "[64Cu]Cu"

"will reach around zero" -> will nearly reach the zero

Otherwise, the referencies are sound and very interesting. The dosimetry part is important, but of my competence, hopefully other reviewers will be complementary.

Comments on the Quality of English Language

Mostly very soundly written,  but maybe my English competence is not enough to criticise this paper.

Author Response

Response to Reviewers

The authors would like to take this opportunity to thank the reviewers for their constructive review and suggestions. These comments have contributed to markedly improving the overall quality of the manuscript. We believe that the various outlined points have been properly discussed and addressed in the manuscript as well as being replied to point-by-point below. If any further concerns remain, we would be happy to address them.

Reviewer #1

Reviewer #1: Thank you for this elaborate review of the latest radiomics and AI methods for pre-therapeutic radiolabeled-SSTR and PSMA PET/CT or PET/MR images for different tasks, such as treatment response prediction, restaging, segmentation, and dosimetry using various analysis methods, including univariate and multivariate statistical analysis and ML approaches. Covering the usefulness of -omics for these treatment techniques is crucial nowadays, as those are quickly developing and important therapeutical options.

Thank you for your careful reading of the manuscript, and we appreciate the positive feedback and compliments on our exploration of radiomics and AI methods in the space of theranostics. We have taken the comments into consideration to improve the manuscript, including considerable edits to improve the manuscript as highlighted by the Yellow color. If any further concerns remain, we would be glad to address them.

From the image analysis point of view, it seemed like the review lacked references cinetic 3D models for pre-treatment dosimetry.

We appreciate your insightful feedback, which has helped us improve and strengthen our study. To address the reviewer’s concern about 3D models for dosimetry, we incorporated the following studies into the sub-section of 3.4. Dose Prediction and Table 2, rows 23 and 24, respectively.

The modification is as follows (Page 27, last paragraph, lines 256-273):

“Xue et al. [1] employed the previously developed DoseGAN software, originally designed for stereotactic body radiation therapy (SBRT), to predict voxel-wise absorbed doses in [177Lu]Lu-PSMA RLT. This prediction took into consideration pre-therapeutic images from [68Ga]Ga-PSMA PET/CT. The authors trained a GAN as a dual-input model based on PET/CT information, utilizing 3D absorbed dose maps acquired through Hermes Voxel Dosimetry (see row 23 in Table 2). To pre-train their GAN model, they further generated 266 digital phantoms using the extended cardiac-torso or XCAT phantom, employing a physiologically based pharmacokinetic (PBPK) model for phantom generation [2]. This approach allowed them to simulate diverse PET images and the spatiotemporal distribution of therapy ligands, ultimately improving dose prediction accuracy (see row 24 in Table 2). In a complementary investigation underscoring the significance of 3D-kinetic models for dosimetry, Kassar et al. [3] explored data augmentation with a PBPK model in a conditional GAN or cGAN to improve organ-specific RLT prediction. Using virtual patient data from realistic simulations, they demonstrated significant enhancements in dose accuracy when incorporating the PBPK constraint. The results suggest that aligning network predictions with mechanistic, patient-specific models addresses limitations in DL for personalized RLT treatment planning.”

From the pharmaceutical point of view, the Table 1 lists all the RPTs that exist, but it is not clear enough:  what is the interest in white lines in the diagnostic part? The fact that only the radio-nuclei are mentionnes and not the pharmaceutics associated, does it mean that all the RPs cited in the diagnostic radio-isotopes pharmaceuticals part were also tested with the therapeutical isotope referred to in the table? Does it mean that for AC225 that was no tests done with Gallium as a theranostic couple? If it is that, there are studies that compared the two in pre-clinical studies: Meyer C, Prasad V, Stuparu A, Kletting P, Glatting G, Miksch J, Solbach C, Lueckerath K, Nyiranshuti L, Zhu S, Czernin J, Beer AJ, Slavik R, Calais J, Dahlbom M. Comparison of PSMA-TO-1 and PSMA-617 labeled with gallium-68, lutetium-177 and actinium-225. EJNMMI Res. 2022 Oct 1;12(1):65. doi: 10.1186/s13550-022-00935-6. PMID: 36182983; PMCID: PMC9526774.

Theranostic Nuclear Medicine with Gallium-68, Lutetium-177, Copper-64/67, Actinium-225, and Lead-212/203 Radionuclides Focus Review Katherine A. Morgan, Stacey E. Rudd, Asif Noor, and Paul S. Donnelly Chemical Reviews 2023 123 (20), 12004-12035 DOI: 10.1021/acs.chemrev.3c00456.  Then I do not understand what this table is clarifying compared to the list mentioned in the text.

We appreciate your valuable comment. We have modified Table 1 accordingly. The new presentation is now more easily to follow as we have merged the white/blank cells to improve clarity. In addition, the focus of this review was on clinical applications so we modified the text and Table caption accordingly to address this issue:

“A complete list of clinically relevant radiotheranostic pairs targeting SSTR and PSMA is shown in Table 1”

“Table 1. Radiotheranostic pairs and targets in NETs and mCRPC diseases, with an emphasis on clinical relevance”

 And small errors :

Table 1, column SSTRs Target/NET, 3rd line: ']' is misplaced.It should be "[64Cu]Cu"

Done. Thanks.

"will reach around zero" -> will nearly reach the zero

Done. Thanks.

Otherwise, the references are sound and very interesting. The dosimetry part is important, but of my competence, hopefully, other reviewers will be complementary.

 We thank again the reviewer for the insightful review and the constructive remarks.

Reviewer 2 Report

Comments and Suggestions for Authors

The article shows an in-depth discussion of the potential risks associated with Radiomics and artificial intelligence in clinical applications. While it emphasizes the potential benefits, it should also address limitations and uncertainties, providing a more comprehensive understanding of the field. I recommend incorporating and emphasizing real-world clinical application cases throughout the article. Introducing concrete examples will provide specific instances supporting the arguments, enhancing the article's practicality and readability. This approach better communicates the actual impact of these technologies on clinical practice.

Comments on the Quality of English Language

Minor editing of English language.

Author Response

The authors would like to take this opportunity to thank the reviewers for their constructive review and suggestions. These comments have contributed to markedly improving the overall quality of the manuscript. We believe that the various outlined points have been properly discussed and addressed in the manuscript as well as being replied to point-by-point below. If any further concerns remain, we would be happy to address them.

Reviewer #2

 The article shows an in-depth discussion of the potential risks associated with Radiomics and artificial intelligence in clinical applications. While it emphasizes the potential benefits, it should also address limitations and uncertainties, providing a more comprehensive understanding of the field. I recommend incorporating and emphasizing real-world clinical application cases throughout the article. Introducing concrete examples will provide specific instances supporting the arguments, enhancing the article's practicality and readability. This approach better communicates the actual impact of these technologies on clinical practice.

 We appreciate the reviewer’s valuable time assessing our work and the positive and constructive remarks. We modified the manuscript as highlighted by the Bright Green color with a new section entitled Clinical Perspectives on Radiomics and AI as follows (Page 36). If any further concerns remain, we would be glad to address them.

“9. Clinical Perspectives on Radiomics and AI

The complexity of patient-specific dosimetry poses challenges for implementation and puts a substantial burden on medical physicists, technologists, and physicians. A prerequisite for personalizing RPTs is not only accuracy and reliability but also practicality. Routine deployment of personalized medicine will become more likely by developments that automate, simplify, or accelerate the dosimetry workflow steps. A key component of personalized RPT in the future will involve use of radiomics and AI methods in the field of radiotheranostics. Personalized therapies can be easily implemented in clinical settings with the assistance of emerging AI research and applications in quantitative imaging, segmentation, absorbed dose estimation, absorbed dose prediction, and outcome modeling. Nevertheless, this transition is not without its challenges, and the implementation of personalized RPTs requires careful consideration of limitations and uncertainties [4].

Currently, patient-specific RPTs are not routinely implemented in clinical settings [5]. In recent years, the emergence of radiomics and AI has provided a means to streamline these tasks, with the possibility of integrating them into clinical practice. In this work, possible applications of radiomics and AI in the clinical radiotheranostics scenario were highlighted. The intention is to inspire the community to broaden and align efforts to achieve routine and reliable RPTs personalization. In this regard, comprehensive validation and fine-tuning on larger patient cohorts are necessary to refine personalized RPT strategies and facilitate clinical translation. Furthermore, radiomics and AI applications in clinical practice remain challenging and remain to emerge. Image reconstruction algorithms, gray-level intensity discretization, and tumor segmentation methods all influence the measurement of quantitative imaging biomarkers [6, 7]. These factors may affect the robustness, repeatability, and reproducibility of the variables and their outcomes. As such, there is a need to increase the robustness of these tools. For instance, the radiomics quality score (RQS) and the imaging biomarker standardization initiative (IBSI) have been introduced to enhance radiomics robustness, offering methodological guidance and standardization for high-throughput image analysis [6, 8]. By improving understanding of radiomic and AI technical aspects, these instruments contribute to gradual harmonization and standardization. With this advancement, radiomics and AI will have more tangible and less hypothetical applications in clinical settings in the future.”

New Added References

[1]         S. Xue, A. Gafita, A. Afshar-Oromieh, M. Eiber, A. Rominger, and K. Shi, "Voxel-wise prediction of post-therapy dosimetry for 177Lu-PSMA I&T therapy using deep learning," ed: Soc Nuclear Med, 2020.

[2]         S. Xue et al., "PBPK pre-trained deep learning for voxel-wise prediction of post-therapy dosimetry for 177Lu-PSMA therapy," ed: Soc Nuclear Med, 2021.

[3]         M. Kassar et al., "PBPK-Adapted Deep Learning for Voxel-Wise Organ Dosimetry Prediction," Nuklearmedizin-NuclearMedicine, vol. 61, no. 02, pp. 178-178, 2022.

[4]         N. Pandit-Taskar et al., "Dosimetry in clinical radiopharmaceutical therapy of cancer: practicality versus perfection in current practice," Journal of Nuclear Medicine, vol. 62, no. Supplement 3, pp. 60S-72S, 2021.

[5]         S. S. James et al., "Current status of radiopharmaceutical therapy," International Journal of Radiation Oncology* Biology* Physics, vol. 109, no. 4, pp. 891-901, 2021.

[6]         P. Lambin et al., "Radiomics: the bridge between medical imaging and personalized medicine," Nature reviews Clinical oncology, vol. 14, no. 12, pp. 749-762, 2017.

[7]         C. Guezennec et al., "Inter-observer and segmentation method variability of textural analysis in pre-therapeutic FDG PET/CT in head and neck cancer," PLoS One, vol. 14, no. 3, p. e0214299, 2019.

[8]         A. Zwanenburg et al., "Results from the image biomarker standardisation initiative," Radiotherapy and Oncology, 2018.